# LEARNING TO REJECT MEETS LONG-TAIL LEARNING

**Harikrishna Narasimhan,     Aditya Krishna Menon,     Wittawat Jitkrittum**
**Neha Gupta**,     **Sanjiv Kumar**
Google Research
{hnarasimhan, adityakmenon, wittawat, nehagup, sanjivk}@google.com

## ABSTRACT

Learning to reject (L2R) is a classical problem where one seeks a classifier capable of abstaining on low-confidence samples. Most prior work on L2R has focused on minimizing the standard misclassification error. However, in many real-world applications, the label distribution is highly imbalanced, necessitating alternate evaluation metrics such as the balanced error or the worst-group error that enforce equitable performance across both the head and tail classes. In this paper, we establish that traditional L2R methods can be grossly sub-optimal for such metrics, and show that this is due to an intricate dependence in the objective between the label costs and the rejector. We then derive the form of the Bayes-optimal classifier and rejector for the balanced error, propose a novel plug-in approach to mimic this solution, and extend our results to general evaluation metrics. Through experiments on benchmark image classification tasks, we show that our approach yields better trade-offs in both the balanced and worst-group error compared to L2R baselines.

## 1 INTRODUCTION

Multi-class classification is typically framed as learning a classifier that can provide accurate predictions on *any* test sample. However, in many real-world problems, it may be preferable to allow the classifier to *abstain* from making a prediction on low-confidence samples. *Learning to reject* (L2R) is the problem of learning such a classifier capable of abstaining on certain samples, at the expense of incurring a fixed cost (Bartlett et al., 2006; El-Yaniv & Wiener, 2010; Cortes et al., 2016; Gangrade et al., 2021; Cortes et al., 2023). Common L2R approaches include confidence-based thresholding or Chow's rule (Chow, 1970), and loss-based strategies which pose the problem as an instance of cost-sensitive learning (Cortes et al., 2016; Ramaswamy et al., 2018; Ni et al., 2019; Mozannar & Sontag, 2020; Charoenphakdee et al., 2021; Verma & Nalisnick, 2022; Mao et al., 2023a).

Most prior studies on L2R have focused on the goal of minimizing the standard misclassification error. However, the misclassification error may not be well-suited to evaluate the performance of classifier when the label distribution is highly imbalanced or *long-tailed* (Buda et al., 2017). In such applications, one often employs evaluation metrics such as the *balanced error* (Brodersen et al., 2010) or the *worst-group error* (Sagawa et al., 2020), which enforce equitable performance across different label groups (e.g., head and tail). This raises a natural, but surprisingly under-explored question: *what is an appropriate L2R mechanism for such generalised evaluation measures?*

In this paper, we present a formal study of the problem of L2R with general evaluation metrics. In the standard classification setup, one typically minimizes a given evaluation measure by formulating an equivalent cost-sensitive problem (Menon et al., 2013; Koyejo et al., 2014). For example, with the balanced error metric, one may re-weight the per-class errors by the *inverse class priors*. Similarly, many commonly used classification metrics can be reduced to minimizing a cost-sensitive error with *fixed* label costs (Narasimhan et al., 2015b). However, applying these reduction techniques to the L2R setup is not straightforward. In particular, even with a metric as simple as the balanced classification error, the resulting label costs are no longer constants, and have an intricate dependence on the rejection mechanism. We characterize the optimal solution for this non-trivial formulation, propose novel plug-in approaches to mimic the solution, and demonstrate their efficacy in experiments.

In summary, we make the following contributions:

Table 1: Summary of Bayes-optimal classifier $h^*$ and rejector $r^*$ for different metrics. We assume a cost $c$ for abstention. With the standard 0-1 error (row 1) and the conditional 0-1 error (row 2), the optimal rejector is a thresholding of the true class-probability of the predicted label. With the balanced error (row 3), the rejector takes a fundamentally different form, and involves *all* class-probabilities, with per-class coefficients $u, v \in \mathbb{R}^K$ that are distribution-dependent. With a general evaluation metric such as the worst-group error (row 4), we have a stochastic combination of multiple classifier-rejector pairs.

| L2R Risk | $h^*(x)$ | $r^*(x)$ | Reference |
|---|---|---|---|
| Standard (1) | $\arg\max_y \eta_y(x)$ | $\max_y \eta_y(x) < 1 - c$ | Chow (1970) |
| Conditional (22) | $\arg\max_y \eta_y(x)$ | $\max_y \eta_y(x) < 1 - c'$ | Lemma 8 (App. C) |
| Balanced (6) | $\arg\max_y u_y \cdot \eta_y(x)$ | $\max_y u_y \cdot \eta_y(x) < \sum_j v_j \cdot \eta_j(x) - c$ | Theorem 1 |
| Generalized (14) | Stochastic combination of $(h, r)$ pairs of the form in row 3 | | Theorem 3 |

(a)  (b)  (c)  (d)

Figure 1: Comparison of balanced error as a function of proportion of samples rejected on the ImageNet-LT dataset. We compare Chow's rule with the proposed plug-in methods. We also include plots of the individual head and tail errors, and the differences between them. We designate classes with 20 or fewer samples in the training set as "tail" classes, and the remaining as "head" classes. The base classifier uses a ResNet-50 architecture. Chow's rule is seen to be substantially better on the head group compared to the tail group. In contrast, the proposed plug-in rules yield significantly lower balanced (worst-group) error. They are also seen to yield a lower gap between the tail and head errors.

(i) We first derive the Bayes-optimal classifier and rejector for the balanced error metric, and show that they have a fundamentally different form from classical baselines such as Chow's rule (Chow, 1970), which reject samples with low prediction confidence (§3, Table 1, Theorem 1).

(ii) We show that vanilla modifications to Chow's rule such as changing the loss used in the base model training are also theoretically sub-optimal for the balanced error. As an alternative, we propose a novel *plug-in* approach that seeks to mimic the Bayes-optimal solution without making any changes to the base model training (§4).

(iii) We then extend our Bayes-optimal characterization to any evaluation metric that can be expressed as a general function of the per-class errors (Theorem 3), and as a concrete example, showcase how to extend our plug-in approach to minimize the worst-group error (§5).

(iv) Through experiments on benchmark long-tailed image classification datasets, we demonstrate that our *plug-in* approach yields significantly better trade-offs than Chow's rule, and is competitive with variants of Chow that require changes to the training process (§6).

## 2 LEARNING TO REJECT AND LONG-TAIL CLASSIFICATION

We begin with some background material. Let $\mathcal{X}$ be an instance space, $\mathcal{Y} = [L] \doteq \{1, 2, \ldots, L\}$ be the label space, and $\mathbb{P}$ be the underlying distribution over $\mathcal{X} \times \mathcal{Y}$. Furthermore, let $\eta_y(x) = \mathbb{P}(y \mid x)$ denote the conditional distribution over labels. Given a training set $S = \{(x_n, y_n)\}_{n \in [N]}$ comprising $N$ i.i.d. draws from $\mathbb{P}$, multi-class classification canonically seeks a *classifier* $h \colon \mathcal{X} \to \mathcal{Y}$ with low *misclassification error* $R_{0\text{-}1}(h) = \mathbb{P}(y \neq h(x))$. We may parameterise $h(x) = \arg\max_{y' \in \mathcal{Y}} f_{y'}(x)$ for a *scorer* $f \colon \mathcal{X} \to \mathbb{R}^L$, which is typically trained by minimizing an empirical *surrogate* risk on $S$.

### 2.1 LEARNING TO REJECT

We are interested in settings where the classifier is provided the option of *abstaining* on select samples. To this end, we consider the goal of learning a classifier $h \colon \mathcal{X} \to \mathcal{Y}$ and a *rejector* $r \colon \mathcal{X} \to \{0, 1\}$. For a given instance $x$, we abstain on a sample whenever $r(x) = 1$, and predict $h(x)$ otherwise.

Prior literature has explored minimizing the standard misclassification error with a constant penalty for abstaining on a sample (Cortes et al., 2016):

$$R_{0\text{-}1}^{\text{rej}}(h, r) = \mathbb{P}\left(y \neq h(x),\, r(x) = 0\right) + c \cdot \mathbb{P}\left(r(x) = 1\right). \tag{1}$$

Intuitively, when $r$ chooses not to abstain, we incur the usual misclassification error; otherwise, we incur the cost $c > 0$. Under mild distributional assumptions, this objective can be equivalently seen as constraining the proportion of abstentions to be within a budget, a formulation often referred to as *selective classification* (Geifman & El-Yaniv, 2017; Gangrade et al., 2021). More precisely, via a Lagrangian analysis (following, e.g., the Neyman-Pearson lemma (Neyman & Pearson, 1933)), for any such abstention budget, there exists an equivalent cost $c$ in Eq. (1); see Appendix B for details.

**Bayes-optimal L2R.** The Bayes-optimal solution for Eq. (1) admits an intuitive form:

$$h^*(x) = \operatorname*{argmax}_{y \in [L]} \eta_y(x); \qquad r^*(x) = 1 \iff \max_{y \in [L]} \eta_y(x) < 1 - c. \tag{2}$$

The classifier abstains whenever the maximum conditional probability falls below a certain threshold, and otherwise, predicts the class with the maximum class probability.

**Post-hoc L2R.** A classical baseline for L2R, often dubbed as Chow's rule (Chow, 1970), seeks to mimic the rejector in Eq. (2). In practice, one may implement Chow's rule by training a scorer $f : \mathcal{X} \to \mathbb{R}^L$ to minimize the softmax cross-entropy (CE) loss:

$$\mathbb{E}_{(x,y) \sim \mathbb{P}}\left[\ell_{\text{ce}}(y, f(x))\right] = \mathbb{E}_{(x,y) \sim \mathbb{P}}\left[\log\left[\textstyle\sum_{y' \in [L]} e^{f_{y'}(x)}\right] - f_y(x)\right]. \tag{3}$$

One then uses the learned scorer $f$ to compute estimates $p_y(x) \propto \exp(f_y(x))$ of the class probabilities $\eta_y(x)$, and constructs the rejector by thresholding the maximum probability $\max_y p_y(x)$ at $1 - c$.

**Loss-based L2R.** As an alternative to Chow's rule, which constructs a rejector post-hoc given an existing classifier, the literature has also explored the design of loss functions that *jointly* learn both the classifier and rejector to optimize the L2R risk (Cortes et al., 2016; Geifman & El-Yaniv, 2017; Ramaswamy et al., 2018; Ni et al., 2019; Mozannar & Sontag, 2020; Charoenphakdee et al., 2021). Of these, Geifman & El-Yaniv minimize a variant of the L2R risk in Eq. (1) where the classification error is conditioned on non-rejected samples. As shown in Appendix C, the optimal rejector for this *conditional* risk has the same form as Eq. (2), but uses a different (distribution-dependent) threshold.

## 2.2 Long-tail Classification

In many real-world applications, the label distribution is highly *class imbalanced* (He & Garcia, 2009), or *long-tailed* (Buda et al., 2017). In such setting, one may wish to evaluate performance separately on minority and majority classes. For example, a common partitioning of the classes is into *head* and *tail* classes, the primary setting of interest in this paper. More generally, let $G_1, \ldots, G_K \subseteq \mathcal{Y}$ denote a paritioning of the labels into $K$ disjoint groups, with $G_i \cap G_j = \emptyset, \forall i \neq j$ and $\cup_{k \in [K]} G_k = \mathcal{Y}$. Let $\pi_k = \mathbb{P}\left(y \in G_k\right)$ denote the proportion of samples belonging to group $G_k$. The misclassification error for label group $G_k$ can then be measured as $\mathbb{P}\left(y \neq h(x) \mid y \in G_k\right)$.

A key metric of interest in this setting is the *balanced* error that computes the average error across label groups (Brodersen et al., 2010; Menon et al., 2013; 2021b):

$$R_{\text{bal}}(h) = \frac{1}{K} \sum_{k \in [K]} \mathbb{P}\left(y \neq h(x) \mid y \in G_k\right) = \frac{1}{K} \sum_{k \in [K]} \frac{1}{\pi_k} \cdot \mathbb{P}\left(y \neq h(x),\, y \in G_k\right). \tag{4}$$

In practice, one may minimize the balanced error by using a balanced version of the cross-entropy loss, implemented, e.g., through adjustments to the model logits (Menon et al., 2021a). The literature also explores a plethora of other techniques to perform well on the balanced error, including data modifications (Yin et al., 2019; Zhang et al., 2019), loss modifications (Cui et al., 2019; Cao et al., 2019; Tan et al., 2020; Jamal et al., 2020; Ren et al., 2020; Wu et al., 2020a; Deng et al., 2021; Kini et al., 2021; Wang et al., 2021), and prediction modifications (Kang et al., 2020; Zhang et al., 2021).

Another evaluation metric of interest is the *worst-group* error, which is typically minimized through distributionally robust optimization (DRO) (Sagawa et al., 2020; Menon et al., 2021b):

$$R_{\text{wst}}(h) = \max_{k \in [K]} \mathbb{P}\left(y \neq h(x) \mid y \in G_k\right) = \max_{k \in [K]} \frac{1}{\pi_k} \cdot \mathbb{P}\left(y \neq h(x),\, y \in G_k\right). \tag{5}$$

Beyond the balanced (worst-group) errors, there are many other metrics popular in the class imbalanced literature, such as the F-measure or the G-mean, which can be generally expressed as functions of the confusion matrix (Menon et al., 2013; Narasimhan et al., 2015b). When learning a stand-alone classifier, minimizing these can be reduced to a cost-sensitive learning problem with *constant* label costs (Narasimhan et al., 2019). We see next that such a simple reduction is not possible when jointly learning a classifier and rejector in an L2R setup, even with a metric as simple as the balanced error.

## 3 LEARNING TO REJECT WITH BALANCED ERROR

In this paper, we provide a formal framework for L2R with general evaluation metrics, beginning in this section with the balanced error metric.

### 3.1 DOES CHOW'S RULE SUFFICE FOR BALANCED ERROR?

The classical Chow's rejection rule described in Eq. (2) can be grossly sub-optimal for the balanced error, and can in fact have an adverse impact on the tail classes. Figure 1(d) provides an illustration of the latter on the long-tailed ImageNet-LT dataset (Liu et al., 2019), where we group all classes with fewer than 20 samples in the training set as the tail classes, and show that Chow's rule is substantially better on the head classes compared to the tail. This suggests that *one cannot naïvely apply* existing L2R methods in settings involving evaluation measures beyond the standard accuracy.

Jones et al. (2021) similarly show in a setting where the groups are defined by spurious attributes that Chow's rule unfairly favors the majority groups. They propose mitigating this by modifying Chow's rule to use a base model trained with DRO. Follow-up work explores the use of a "sufficiency" constraint in the base model training to alleviate group disparities (Lee et al., 2021). However, formal understanding of L2R with the balanced error remains elusive; we aim to resolve this in the sequel.

### 3.2 FORMULATION WITH REJECTOR-DEPENDENT COSTS

We now formally describe the problem of L2R with the balanced error. For a base classifier $h$ and a rejector $r$, we define the error that the classifier makes on non-abstained samples from each group by:

$$\mathbb{P}\left(y \neq h(x) \,|\, r(x) = 0, \, y \in G_k\right),$$

where we condition on samples from group $G_k$ which the rejector does *not* reject (Jones et al., 2021). We consider the goal of minimizing the average per-group error, with a constraint on the total proportion of abstentions. One may equivalently impose a a rejection penalty of $c > 0$:

$$R_{\text{bal}}^{\text{rej}}(h, r) = \frac{1}{K} \sum_{k \in [K]} \mathbb{P}\big(y \neq h(x) \mid r(x) = 0, \, y \in G_k\big) + c \cdot \mathbb{P}\left(r(x) = 1\right). \tag{6}$$

We will refer to this objective as the *balanced L2R risk*, which we re-write as:

$$R_{\text{bal}}^{\text{rej}}(h, r) = \frac{1}{K} \sum_{k \in [K]} \frac{1}{\pi_k(r)} \cdot \mathbb{P}\left(y \neq h(x), \, r(x) = 0, \, y \in G_k\right) + c \cdot \mathbb{P}\left(r(x) = 1\right).$$

where $\pi_k(r) = \mathbb{P}\left(r(x) = 0, \, y \in G_k\right)$ is the proportion of samples from group $G_k$ that were *not* rejected by $r$. Similar to the balanced error in Eq. (4), each group is weighted by the inverse group prior; however, the coefficients here are not constants and are intricately tied to the rejector $r$. In fact, the above risk has a *non-decomposable* dependence on $r$: both the numerator and the denominator depend on $r$, making the minimization problem non-trivial and challenging. This is in contrast to prior work on cost-sensitive rejection (deferral), where the costs on the misclassification errors are *fixed* and have no dependence on $r$ (Santos-Pereira & Pires, 2005; Alves et al., 2024).

### 3.3 BAYES-OPTIMAL CLASSIFIER AND REJECTOR

As a first step towards learning a classifier and rejector that minimizes the balanced L2R risk in Eq. (6), we seek to understand the form of the Bayes-optimal solution for the objective. To this end, we re-write the risk into an equivalent form where the denominators do not explicitly depend

on the rejector. Specifically, we decouple the dependence of the denominators on the rejector by introducing auxiliary variable $\alpha_1, \ldots, \alpha_K \in (0, 1)$, and reformulate the minimization over $h$ and $r$ as an equivalent constrained optimization problem:

$$\min_{h, r, \alpha \in (0,1)^K} \sum_{k \in [K]} \frac{1}{\alpha_k} \cdot \mathbb{P}\left(y \neq h(x), r(x) = 0, y \in G_k\right) + c \cdot \mathbb{P}\left(r(x) = 1\right) \tag{7}$$

$$\text{s.t.} \quad K \cdot \mathbb{P}\left(r(x) = 0, y \in G_k\right) = \alpha_k, \quad \forall k \in [K].$$

This then allows us to provide a characterization of the Bayes-optimal solution. By computing the Lagrangian for Eq. (7) with multipliers $\mu_1, \ldots, \mu_K \in \mathbb{R}$ for the $K$ equality constraints, we have:

**Assumption 1.** For $x \sim \mathbb{P}$, assume that $[\eta_1(x), \ldots, \eta_L(x)]$ is a continuous random vector.

**Assumption 2.** The minimizer $(h^*, r^*)$ of the balanced L2R risk in Eq. (6) is such that $\mathbb{P}(r^*(x) = 0, y \in G_k) \geq \kappa, \forall k$, for some $\kappa > 0$.

**Theorem 1.** *Under Assumptions 1 and 2, there exists group-specific parameters $\alpha^* \in (0, K)^K$ and $\mu^* \in \mathbb{R}^K$ such that the optimal classifier and rejector for Eq. (6) take the form:*

$$h^*(x) = \arg\max_{y \in [L]} \frac{1}{\alpha^*_{[y]}} \cdot \eta_y(x);$$

$$r^*(x) = 1 \iff \max_{y \in [L]} \frac{1}{\alpha^*_{[y]}} \cdot \eta_y(x) < \sum_{y' \in [L]} \left(\frac{1}{\alpha^*_{[y']}} - \mu^*_{[y']}\right) \cdot \eta_{y'}(x) - c, \tag{8}$$

*where $[y]$ is the index of the group class $y$ belongs to; moreover, $\alpha^*_k = K \cdot \mathbb{P}\left(r^*(x) = 0, y \in G_k\right)$.*

Assumption 1 is fairly common in the analysis of Bayes-optimal solutions for evaluation metrics (Narasimhan et al., 2015b; Yan et al., 2018; Yang et al., 2020). This together with Assumption 2 allows us to pose Eq. (7) as an equivalent (unconstrained) Lagrangian minimization problem.

The optimal rejector uses the term $\max_{y \in [L]} \frac{1}{\alpha^*_{[y]}} \cdot \eta_y(x)$ as a measure of classifier confidence, and checks if it falls below a *sample-dependent* threshold. The multipliers $\mu^*_k$s in the threshold can be seen as per-group adjustments that ensure that the acceptance rate for group $k$ is proportional to $\alpha^*_k$. In fact, in the special case where we have binary labels, each in a separate group, the rejector takes a much simpler form, with a constant threshold for each class (details in Appendix D.4).

**Remark 1** (**Sub-optimality of L2R Baselines**). Unlike the classical Chow's rule in Eq. (2) or its conditional variant in Eq. (23), which apply a constant threshold to the maximum conditional probability $\max_y \eta_y(x)$, the optimal rejector in Theorem 1 takes a fundamentally different form, involving *all* class-probabilities. In fact, it can be seen as applying a *sample-dependent* threshold to re-weighted probabilities. This suggests that Chow's rule is sub-optimal for the balanced L2R risk. This also suggests that approaches which optimize a surrogate loss for the L2R risk (Cortes et al., 2016; Geifman & El-Yaniv, 2017; Ramaswamy et al., 2018; Ni et al., 2019; Mozannar & Sontag, 2020; Charoenphakdee et al., 2021) are also sub-optimal for the balanced L2R risk, as they are also designed to learn the same thresholded rejector in Eq. (2) that Chow's rule seeks to mimic.

## 4 ALGORITHMS FOR BALANCED L2R

We next present an algorithm for learning a classifier and rejector that is optimal for the balanced L2R risk in Eq. (6). We begin by exploring simple modifications to Chow's rule.

### 4.1 CHOW'S RULE WITH MODIFIED BASE LOSS

One obvious variation to Chow's rule is to modify the base model training to optimize a balanced version of the CE loss in Eq. (3):

$$f^{\text{bal}} = \text{argmin}_f \sum_{k \in [K]} \frac{1}{\pi_k} \cdot \mathbb{E}\left[\mathbf{1}(y \in G_k) \cdot \ell_{\text{ce}}(y, f(x))\right]. \tag{9}$$

One may then implement Chow's rule via the class probability estimates $p^{\text{bal}}_y(x) \propto \exp(f^{\text{bal}}_y(x))$:

$$h^{\text{bal}}(x) \in \arg\max_{y \in [L]} p^{\text{bal}}_y(x); \quad r^{\text{bal}}(x) = 1 \iff \max_{y \in [L]} p^{\text{bal}}_y(x) < 1 - c. \tag{10}$$

Although intuitive, this simple modification to Chow's rule does not yield the optimal classifier and rejector for the balanced L2R risk.

**Lemma 2.** *The variant of Chow's rule in Eq. (10), where the base model is trained with the balanced cross-entropy loss, results in a classifier and rejector of the form:*

$$h^{\mathrm{bal}}(x) = \arg\max_{y\in[L]} \frac{1}{\pi_{[y]}} \cdot \eta_y(x)$$

$$r^{\mathrm{bal}}(x) = 1 \iff \max_{y\in[L]} \frac{1}{\pi_{[y]}} \cdot \eta_y(x) < \Big( \sum_{y'\in[L]} \frac{1}{\pi_{[y']}} \cdot \eta_{y'}(x) \Big) \cdot (1-c), \qquad (11)$$

*where $[y]$ is the index of the group to which class $y$ belongs.*

There are two key differences between the above rejector and the optimal rejector in Theorem 1. First, in the above rejector, the abstention cost $c$ is part of a *multiplicative* term whereas it is part of an *additive* term in the rejector in Theorem 1. Second, the per-class weights in the above rejector depend on the fixed group priors $\pi_k$, whereas those in Theorem 1 depend more intricately on the underlying distribution through the parameters $\alpha_k^*$ (which depend on the optimal rejector) and multipliers $\mu_k^*$. More generally, any re-weighting of the base loss with constant coefficients will result in a rejector with a multiplicative penalty term different from the form in Theorem 1 (see Appendix A.2).

### 4.2 PLUG-IN APPROACH WITH NO CHANGES TO THE BASE LOSS

As an alternative to modifying the base loss in Chow's rule, we propose a new plug-in approach that seeks to directly mimic the optimal classifier and rejector for the balanced L2R risk. Like standard Chow's rule, we assume access to only a class probability estimator $p_y(x)$ trained to minimize the softmax CE loss in Eq. (3). We then wish to approximate the Bayes-optimal classifier and rejector from Theorem 1 by plugging in estimates $p_y(x)$ of the class probabilities:

$$\hat{h}(x) = \arg\max_{y\in[L]} \frac{1}{\hat{\alpha}_{[y]}} \cdot p_y(x); \qquad (12)$$

$$\hat{r}(x) = 1 \iff \max_{y\in[L]} \frac{1}{\hat{\alpha}_{[y]}} \cdot p_y(x) < \sum_{y'\in[L]} \Big( \frac{1}{\hat{\alpha}_{[y']}} - \hat{\mu}_{[y']} \Big) \cdot p_{y'}(x) - c, \qquad (13)$$

for choices of $\hat{\alpha} \in (0, K)^K$ and $\hat{\mu} \in \mathbb{R}^K$. One may carry out an exhaustive search to pick $\hat{\alpha}$ and $\hat{\mu}$ so that the resulting classifier-rejector pair yields the lowest balanced error on a held-out sample. However, searching over $2K$ parameters can quickly become infeasible even when $K$ is small.

To make the search more efficient, we exploit the additional constraint in Theorem 1 that requires $\hat{\alpha}_k$ to match the rejector's coverage for group $G_k$, i.e., $\hat{\alpha}_k = K \cdot \mathbb{P}(y \in G_k, \hat{r}(x) = 0)$. Specifically, we propose applying a grid search over only the multipliers $\hat{\mu}$, while using a power-iteration-like heuristic to choose $\hat{\alpha}$. Starting with initial values $\alpha^{(0)} \in (0, K)^K$, we propose alternating between constructing the classifier-rejector pair and updating the group priors using the rejector's coverage:

- Construct $(h^{(m)}, r^{(m)})$ using equations 12–13 with $\hat{\alpha} = \alpha^{(m)}$ and the fixed multipliers $\hat{\mu}$;
- Set $\alpha_k^{(m+1)} = K \cdot \mathbb{P}(y \in G_k, r^{(m)}(x) = 0)$.

One may repeat these steps until the iterates $\alpha^{(t)}$s converge to a stationary point, and return the corresponding classifier and rejector.

For moderate number of groups $K$, one may apply the above procedure for different values of multipliers $\hat{\mu} \in \mathbb{R}^K$, and pick the one for which the returned classifier-rejector pair yields the lowest balanced error on a validation set. The details are outlined in Algorithm 1, where $\Lambda$ is the set of multiplier values we wish to search over, and $\beta_k = \frac{1}{K}$ for the balanced error. In fact, one can further prune the search space by re-parameterizing the rejection criterion in Eq. (13) to search over only $K - 1$ parameters (see Appendix E.1 for details). For problems with two groups (e.g. head and tail), this would mean that we only have a *single* parameter to tune. For larger number of groups, we prescribe applying an alternate Lagrangian-style min-max optimization to tune the multiplier parameters. We elaborate on this procedure in Appendix E.2.

---

**Algorithm 1** Cost-sensitive Plug-in (CS-plug-in)

---

1: **Input:** Rejection cost $c$, Group weights $\beta \in \Delta_L$, Pre-trained model $p : \mathcal{X} \to \Delta_L$, Sample $S$
2: **Parameters:** Multiplier set $\Lambda \subset \mathbb{R}^K$, Iterations $M$, Initial prior $\alpha^{(0)} \in (0,1)^K$, Initial $(\hat{h}, \hat{r})$
3: **For** $\mu$ in $\Lambda$:
4:     **For** $m = 0$ to $M - 1$:
5:         Construct $(h^{(m+1)}, r^{(m+1)})$ using equations 12–13 with $\hat{\alpha}_k = \frac{\alpha_k^{(m)}}{\beta_k}$ and $\hat{\mu} = \mu$
6:         $\alpha_k^{(m+1)} = \frac{1}{|S|} \sum_{(x,y) \in S} \mathbf{1}(y \in G_k, r^{(m+1)}(x) = 0), \ \forall k \in [K]$
7:     $(\hat{h}, \hat{r}) = (h^{(M)}, r^{(M)})$ if $\sum_k \beta_k \cdot \hat{e}_k(h^{(M)}, r^{(M)}) < \sum_k \beta_k \cdot \hat{e}_k(\hat{h}, \hat{r})$
8: **Return:** $(\hat{h}, \hat{r})$

---

**Algorithm 2** Worst-group Plug-in

---

1: **Input:** Rejection cost $c$, Pre-trained model $p : \mathcal{X} \to \Delta_L$, Sample $S^{\mathrm{val}}$ divided into $S_1, S_2$
2: **Parameters:** Inital group weights $\beta^{(0)} \in \Delta_L$, Iterations $T$, Step-size $\xi$
3: **For** $t = 0$ to $T$
4:     $(h^{(t)}, r^{(t)}) = \text{CS-plug-in}(c, \beta^{(t)}, p, S_1)$
5:     $\beta_k^{(t+1)} = \frac{1}{Z} \cdot \beta_k^{(t)} \cdot \exp(\xi \cdot \hat{e}_k(h^{(t)}, r^{(t)})), \ \forall k \in [K]$, where $Z = \sum_j \beta_j^{(t)} \cdot \exp(\xi \cdot \hat{e}_j(h^{(t)}, r^{(t)}))$
6: **Return:** $(h^{(T)}, r^{(T)})$

---

Figure 2: Plug-in algorithm for the cost-sensitive error $\sum_k \beta_k \cdot \hat{e}_k(h, r)$ (Algorithm 1) and worst-group error $\max_k \hat{e}_k(h, r)$ (Algorithm 2), where $\hat{e}_k(h, r) = \frac{\sum_{(x,y) \in S_2} \mathbf{1}(y \neq h(x), y \in G_k, r(x) = 0)}{\sum_{(x,y) \in S_2} \mathbf{1}(y \in G_k, r(x) = 0)}$. Algorithm 1 can be applied to minimize the balanced L2R risk in Eq. (6) by setting $\beta_k = 1/K, \forall k \in [K]$.

## 5   Extension to General Evaluation Metrics

We next extend our results to evaluation metrics more general than the balanced error. To this end, we denote the conditional error of classifier $h$ and rejector $r$ on group $G_k$ by: $e_k(h, r) = \mathbb{P}(y \neq h(x) \mid r(x) = 0, y \in G_k)$. We then consider the problem of minimizing a general function $\psi : [0, 1]^K \to \mathbb{R}$ of the per-group errors $e_1(h, r), \ldots, e_K(h, r)$, with a cost of $c > 0$ for abstention:

$$R_{\mathrm{gen}}^{\mathrm{rej}}(h, r) = \psi\left(e_1(h, r), \ldots, e_K(h, r)\right) + c \cdot \mathbb{P}(r(x) = 1). \tag{14}$$

This *generalized L2R* risk includes the balanced error from Eq. (6) as a special case when $\psi(\mathbf{z}) = \frac{1}{K} \sum_k z_k$. It also captures more general evaluation metrics defined by a *non-linear* function $\psi$ such as the worst-group error with $\psi(\mathbf{z}) = \max_{k \in [K]} z_k$ (Sagawa et al., 2020), and the G-mean metric when $\psi(\mathbf{z}) = 1 - (\prod_{k=1}^{K}(1 - z_k))^{1/K}$ (Menon et al., 2013; Narasimhan et al., 2015b).

### 5.1   Bayes-optimal Classifier and Rejector

As with prior work on optimizing general evaluation metrics (Narasimhan et al., 2015a; Yang et al., 2020), the generalized L2R risk is minimized by a classifier-rejector mechanism that is stochastic.

**Theorem 3.** *For any non-linear $\psi$, there exists coefficients $u^{(1)}, v^{(1)}, \ldots, u^{(K+1)}, v^{(K+1)} \in \mathbb{R}^K$, and a distribution $\nu \in \Delta_{K+1}$, such that, the generalized L2R risk in Eq. (14) is minimized by a stochastic mechanism that randomizes over $K + 1$ classifier-rejector pairs: for any $x$, it predicts using $(h^{(i)}, r^{(i)})$ with probability $\nu_i$:*

$$h^{(i)}(x) \in \arg\max_y u_y^{(i)} \cdot \eta_y(x), \quad i \in [K+1]$$

$$r^{(i)}(x) = 1 \iff \max_y u_y^{(i)} \cdot \eta_y(x) < \sum_{y'} v_{y'}^{(i)} \cdot \eta_{y'}(x) - c, \quad i \in [K+1].$$

Stochastic classifiers are widely used in the design of learning algorithms for complex (non-linear) evaluation metrics (Narasimhan et al., 2015b; Cotter et al., 2019b; Narasimhan et al., 2019; Yang et al., 2020). In applications where the use of stochastic classifiers is infeasible, one may use techniques offered in Cotter et al. (2019a) to convert them into a similar performing deterministic classifier.

| Method | CIFAR-100 | | ImageNet | | iNaturalist | |
|---|---|---|---|---|---|---|
| | Balanced | Worst | Balanced | Worst | Balanced | Worst |
| Chow | $0.509 \pm 0.002$ | $0.883 \pm 0.007$ | $0.409 \pm 0.003$ | $0.685 \pm 0.007$ | $0.131 \pm 0.004$ | $0.162 \pm 0.004$ |
| CSS | $0.483 \pm 0.002$ | $0.785 \pm 0.003$ | $0.526 \pm 0.002$ | $0.784 \pm 0.003$ | $0.200 \pm 0.005$ | $0.246 \pm 0.006$ |
| Chow [BCE] | $0.359 \pm 0.017$ | $0.570 \pm 0.030$ | $0.213 \pm 0.001$ | $0.274 \pm 0.003$ | $0.111 \pm 0.003$ | $\mathbf{0.114 \pm 0.003}$ |
| Chow [DRO] | $0.325 \pm 0.004$ | $0.333 \pm 0.003$ | $\mathbf{0.197 \pm 0.001}$ | $\mathbf{0.211 \pm 0.004}$ | $\mathbf{0.110 \pm 0.001}$ | $0.117 \pm 0.002$ |
| Plug-in [Balanced] | $0.292 \pm 0.006$ | $0.416 \pm 0.015$ | $0.205 \pm 0.001$ | $0.248 \pm 0.001$ | $\mathbf{0.108 \pm 0.003}$ | $0.118 \pm 0.004$ |
| Plug-in [Worst] | $\mathbf{0.287 \pm 0.008}$ | $\mathbf{0.321 \pm 0.018}$ | $\mathbf{0.198 \pm 0.001}$ | $\mathbf{0.213 \pm 0.004}$ | $\mathbf{0.108 \pm 0.001}$ | $\mathbf{0.113 \pm 0.002}$ |

Table 2: Area Under the Risk-Coverage curve (AURC) with the balanced and worst-group error. Chow and CSS are L2R baselines. We evaluate two variants of Chow (BCE, DRO) with a modified base loss (§4.1), and plug-in methods (Balanced, Worst) that require *no changes to the base model* training (§4.2, §5).

## 5.2 MINIMIZING WORST-GROUP ERROR

One can minimize the risk in Eq. (14) by exploiting the form of the function $\psi$. For example, if $\psi$ is *convex*, one can reduce it to a sequence of minimization problems of the form:

$$\sum_k \beta_k^{(t)} \cdot e_k(h, r) + c \cdot \mathbb{P}\left(r(x) = 1\right),$$

for coefficients $\beta^{(1)}, \beta^{(2)}, \ldots \in \mathbb{R}_+^K$, each of which can be solved by adapting the plug-in method described in §4.2. Prior work has shown how this can be done in the standard classification setup (without rejection) by applying gradient-based optimization to $\psi$ (Narasimhan et al., 2019; 2022a; Yang et al., 2020). As a concrete example, we showcase how a similar reduction can be done for the worst-group L2R risk with $\psi(z_1, \ldots, z_K) = \max_{k \in [K]} z_k$. The details are in Algorithm 2.

The algorithm maintains group-specific weights $\beta^{(t)} \in \Delta_K$, where $\Delta_K$ is the probability simplex over $K$ coordinates. At each iteration, it updates $\beta^{(t)}$ using exponentiated gradient-descent, up-weighting groups with high empirical error $\hat{e}_k(h, r)$. It then invokes the plug-in approach in Algorithm 1 to minimize the resulting weighted error $\sum_k \beta_k^{(t)} \cdot \hat{e}_k(h, r)$. Below, we provide convergence guarantees for the algorithm in terms of the optimality gap in the weighted error minimization step (line 4).

**Theorem 4.** *Let $R_{\text{wst}}^{\text{rej}}(h, r)$ denote the generalized L2R risk in Eq. (14) with $\psi(z_1, \ldots, z_K) = \max_k z_k$. Let $\epsilon_t^{\text{gen}} = |e_k(h^{(t)}, r^{(t)}) - \hat{e}_k(h^{(t)}, r^{(t)})|$ denote the estimation error in iteration $t$ of Algorithm 2. Let $\epsilon_t^{\text{cs}} = \sum_k \beta_k^{(t)} \cdot e_k(h^{(t)}, r^{(t)}) - \inf_{h,r} \sum_k \beta_k^{(t)} \cdot e_k(h, r)$ denote the excess cost-sensitive risk for the classifier-rejector pair $(h^{(t)}, r^{(t)})$ returned in line 4. Then a stochastic mechanism that picks one among $\{(h^{(1)}, r^{(1)}), \ldots, (h^{(T)}, r^{(T)})\}$ with equal probability satisfies:*

$$\max_{k \in [K]} \mathbb{E}_t\left[e_k(h^{(t)}, r^{(t)})\right] + c \cdot \mathbb{E}_t\left[\mathbb{P}(r^{(t)}(x) = 1)\right] \leq \inf_{h,r} R_{\text{wst}}^{\text{rej}}(h, r) + \bar{\epsilon}^{\text{cs}} + 2\bar{\epsilon}^{\text{gen}} + 2\sqrt{\frac{\log(K)}{T}},$$

*where $t$ is drawn uniformly from $\{1, \ldots, T\}$, $\bar{\epsilon}^{\text{cs}} = \frac{1}{T} \sum_{t \in [T]} \epsilon_t^{\text{cs}}$ and $\bar{\epsilon}^{\text{gen}} = \frac{1}{T} \sum_{t \in [T]} \epsilon_t^{\text{gen}}$.*

To ensure a fair comparison against other deterministic baselines, when implementing Algorithm 2, we do not employ a stochastic solution, but instead use the final classifier-rejector pair $(h^{(T)}, r^{(T)})$.

## 6 EXPERIMENTAL RESULTS

We present experiments on long-tailed image classification tasks to showcase that proposed plug-in approaches for the balanced error (§4) and the worst-group error (§5) yield significantly better trade-offs than Chow's rule, despite using the same base model, and are competitive with variants of Chow's rule which require re-training the base model with a modified loss.

**Datasets.** We replicate the long-tail experimental setup from Menon et al. (2021a). We use long-tailed versions of CIFAR-100 (Krizhevsky, 2009), ImageNet (Deng et al., 2009) and iNaturalist (Van Horn et al., 2018). For CIFAR-100, we downsample the examples per label following the Exp profile of Cao et al. (2019), and for ImageNet, we use the long-tail version provided by Liu et al. (2019). The train, test and validation splits have the same label distributions (see Appendix F for details). We train a ResNet-32 (50) model for CIFAR (ImageNet and iNaturalist).

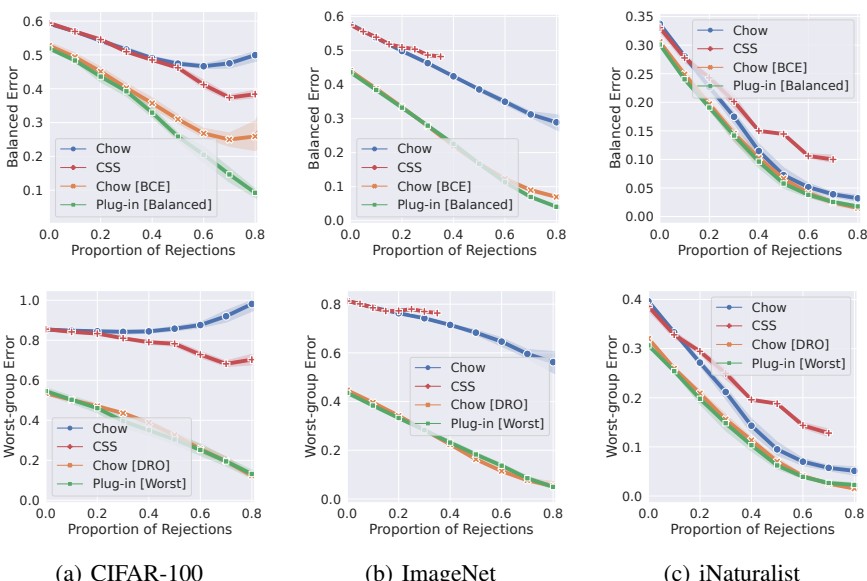

Figure 3: Balanced and worst-group errors as functions of proportion of rejections. *Lower is better*.

**Evaluation metrics.** For each dataset, we divide the classes into two groups, namely *head* and *tail*. We designate all classes with 20 or fewer samples in the training set as tail classes (Liu et al., 2019), and the remaining as the head. We evaluate both the *balanced error* and the *worst-group* error across the head and tail groups. We train classifiers and rejectors with different rejection costs $c$, and plot these metrics as a function of the proportion of rejections. We summarize the performance across different rejection rates by computing the areas under the respective curves (averaged over *5 trials*); this summary metric is referred to as the area under risk-coverage curve (AURC) (Moon et al., 2020).

**Comparisons.** We compare against two representative L2R baselines that optimize the standard 0-1 error: (i) Chow's rule, which trains a base model by optimizing the CE loss, and thresholds the estimated max probability (Eq. (2)); and (ii) the cost-sensitive softmax (CSS) loss of Mozannar & Sontag (2020), which jointly trains a classifier and rejector using a surrogate loss (with the ResNet model predicting $L + 1$ logits, and the $(L + 1)$-th denoting the 'abstain' option). We evaluate our plug-in methods for the balanced error (Alg. 1) and the worst-case error (Alg. 2), both of which use the same base model as Chow's rule. We also evaluate variants of Chow's rule (§4.1), where the base model is trained with either a balanced cross-entropy (BCE) loss implemented through logit adjustments (Menon et al., 2021a), or the DRO loss prescribed by Jones et al. (2021).

**Results.** We present plots of the balanced and worst-group errors as a function of the rejection rate (for varying values of $c$) in Figure 3. We summarize the area under these curves in Table 2. On all three datasets, the proposed plug-in methods are substantially better than both Chow's rule and the CSS baseline. The plug-in methods are also comparable to or better than the variants of Chow's rule that uses a balanced or DRO base loss. As shown in Figures 4–5 in Appendix F.5, Chow's rule is unsurprisingly the best on the standard 0-1 error. On the other hand, Figures 6–7 show that our plug-in methods yield a lower gap between the tail and head errors compared to the L2R baselines. Finally, in Appendix F.6, we show on CIFAR-100 that the heuristic we employ in our plug-in approach to pick the group coverage $\alpha$ (§4.2) performs comparable to an exhaustive grid search over $\alpha$.

## 7 CONCLUSION

We have shown that standard L2R can be sub-optimal for general evaluation metrics, and proposed simple plug-in methods that yield significantly better trade-offs for both the balanced and worst-group error. Our methods require only a pre-trained model optimized with the CE loss, and are competitive with complex loss modification approaches. In the future, it is of interest to explore joint training of the classifier and rejector using surrogate losses (Cao et al., 2022), and extensions to hierarchical classification (Wu et al., 2020b) and pairwise ranking (Shen et al., 2020; Mao et al., 2023b).

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

# Appendix

## A  PROOFS FOR RESULTS IN MAIN TEXT

### A.1  PROOF OF THEOREM 1

The proof follows as a corollary of Theorem 12 in Appendix D.3 by setting $\beta_k = \frac{1}{K}, \forall k$, and scaling the resulting parameters $\alpha_y^*$ by $K$. Note that Theorem 12 considers a minimization problem that allows for stochastic classifier-rejector mechanisms. However, since the optimal solution is a deterministic classifier-rejector pair, it still remains the optimal solution when the minimization is over only deterministic classifier-rejector pairs.

### A.2  PROOF OF LEMMA 2

We prove a more general version of Lemma 2. Consider a variation of Chow's rule where we modify the base model training to optimize a weighted version of the CE loss in Eq. (3):

$$\sum_{k \in [K]} \beta_k \cdot \mathbb{E}\left[\mathbf{1}(y \in G_k) \cdot \ell_{\mathrm{ce}}(y, f(x))\right], \tag{15}$$

for some $\beta \in \mathbb{R}_+^K$. One may then use the class probability estimates $p_y^{\mathrm{wt}}(x) \propto \exp(f_y^{\mathrm{wt}}(x))$ from the learned scorer $f^{\mathrm{wt}}$ to implement Chow's rule:

$$h^{\mathrm{wt}}(x) \in \arg\max_{y \in [L]} p_y^{\mathrm{wt}}(x); \qquad r^{\mathrm{wt}}(x) = 1 \iff \max_{y \in [L]} p_y^{\mathrm{wt}}(x) < 1 - c. \tag{16}$$

**Lemma 5.** *The variant of Chow's rule in Eq. (16), where the base model is trained with the balanced cross-entropy loss, results in a classifier and rejector of the form:*

$$h^{\mathrm{wt}}(x) = \arg\max_{y \in [L]} \beta_{[y]} \cdot \eta_y(x)$$

$$r^{\mathrm{bal}}(x) = 1 \iff \max_{y \in [L]} \beta_{[y]} \cdot \eta_y(x) < \left(\sum_{y' \in [L]} \beta_{[y']} \cdot \eta_{y'}(x)\right) \cdot (1 - c), \tag{17}$$

*where $[y]$ is the index of the group to which class $y$ belongs.*

*Proof.* We re-write the weighted CE loss in Eq. (15) as:

$$\sum_{y \in [L]} \mathbb{E}\left[\beta_{[y]} \cdot \ell_{\mathrm{ce}}(y, f(x))\right].$$

Since the softmax CE loss is a strictly proper-composite loss, we have that the minimizer $p_y^{\mathrm{wt}}(x) \propto \exp(f_y^{\mathrm{wt}}(x))$ of this objective takes the form:

$$p_y^{\mathrm{wt}}(x) = \frac{1}{Z(x)} \cdot \beta_{[y]} \cdot \eta_y(x),$$

where $Z(x) = \sum_{y'} \beta_{[y']} \cdot \eta_{y'}(x)$. When applying Chow's rule to these probabilities, we get:

$$r^{\mathrm{wt}}(x) = 1 \iff \max_y p_y^{\mathrm{wt}}(x) < 1 - c,$$

which is the same as:

$$r^{\mathrm{wt}}(x) = 1 \iff \max_y \beta_{[y]} \cdot \eta_y(x) < \left(\sum_{y'} \beta_{[y']} \cdot \eta_{y'}(x)\right) \cdot (1 - c),$$

as desired. □

*Proof of Lemma 2.* The proof follows from Lemma 5 by setting $\beta_k = \frac{1}{\pi_k}$. □

### A.3 PROOF OF THEOREM 3

See Appendix D.5.

### A.4 PROOF OF THEOREM 4

The proof follows from an application of the standard convergence guarantees for the exponentiated gradient descent algorithm (Shalev-Shwartz et al., 2012). One can further bound the estimation errors $\epsilon_t^{\text{gen}}$ in each iteration of Algorithm 2 using standard generalization analysis (Narasimhan et al., 2015a; Tavker et al., 2020; Wang et al., 2023), noting that the classifier and rejectors $(h^t, r^t)$ are plug-in estimators chosen from a finite capacity class (namely that of post-hoc linear adjustments to a fixed model $p$).

*Proof.* Recall that the worst-group L2R risk is given by:

$$R_{\text{wst}}^{\text{rej}}(h, r) = \max_k e_k(h, r) + c \cdot \mathbb{P}(r(x) = 1).$$

This risk can be equivalently re-written as:

$$R_{\text{wst}}^{\text{rej}}(h, r) = \max_{\beta \in \Delta_K} \sum_k \beta_k \cdot e_k(h, r) + c \cdot \mathbb{P}(r(x) = 1).$$

We apply standard guarantees for the exponentiated gradient ascent algorithm (Shalev-Shwartz et al., 2012) to treating $\hat{e}_1(h^{(t)}, r^{(t)}), \ldots, \hat{e}_K(h^{(t)}, r^{(t)})$ as the losses incurred by $K$ experts. We have that when $\xi = \sqrt{\frac{\log(K)}{T}}$, the iterates $(h^{(1)}, r^{(1)}), \ldots, (h^{(T)}, r^{(T)})$ in Algorithm 2 satisfy:

$$\max_{\beta \in \Delta_K} \frac{1}{T} \sum_{t=1}^T \sum_k \beta_k \cdot \hat{e}_k(h^{(t)}, r^{(t)}) \le \frac{1}{T} \sum_{t=1}^T \sum_k \beta_k^{(t)} \cdot \hat{e}_k(h^{(t)}, r^{(t)}) + 2\sqrt{\frac{\log(K)}{T}}.$$

Bounding by the estimation error in each iteration, we get:

$$\max_{\beta \in \Delta_K} \frac{1}{T} \sum_{t=1}^T \sum_k \beta_k \cdot (e_k(h^{(t)}, r^{(t)}) - \epsilon_t^{\text{gen}})$$

$$\le \frac{1}{T} \sum_{t=1}^T \sum_k \beta_k^{(t)} \cdot (e_k(h^{(t)}, r^{(t)}) + \epsilon_t^{\text{gen}}) + 2\sqrt{\frac{\log(K)}{T}},$$

which is the same as:

$$\max_{\beta \in \Delta_K} \frac{1}{T} \sum_{t=1}^T \sum_k \beta_k \cdot e_k(h^{(t)}, r^{(t)}) \le \frac{1}{T} \sum_{t=1}^T \sum_k \beta_k^{(t)} \cdot e_k(h^{(t)}, r^{(t)}) + 2\bar{\epsilon}^{\text{gen}} + 2\sqrt{\frac{\log(K)}{T}}.$$

Adding $c \cdot \mathbb{P}(r^{(t)}(x) = 1)$ on either side, we have:

$$\max_{\beta \in \Delta_K} \frac{1}{T} \sum_{t=1}^T \sum_k \beta_k \cdot e_k(h^{(t)}, r^{(t)}) + c \cdot \mathbb{P}(r^{(t)}(x) = 1)$$

$$\le \frac{1}{T} \sum_{t=1}^T \sum_k \beta_k^{(t)} \cdot e_k(h^{(t)}, r^{(t)}) + c \cdot \mathbb{P}(r^{(t)}(x) = 1) + 2\bar{\epsilon}^{\text{gen}} + 2\sqrt{\frac{\log(K)}{T}}$$

$$\le \frac{1}{T} \sum_{t=1}^T \inf_{h,r} \left\{ \sum_k \beta_k^{(t)} \cdot e_k(h, r) + c \cdot \mathbb{P}(r(x) = 1) \right\} + \frac{1}{T} \sum_t \epsilon_t^{\text{cs}} + 2\bar{\epsilon}^{\text{gen}} + 2\sqrt{\frac{\log(K)}{T}} \quad (18)$$

$$\le \inf_{h,r} \frac{1}{T} \sum_{t=1}^T \sum_k \beta_k^{(t)} \cdot e_k(h, r) + c \cdot \mathbb{P}(r(x) = 1) + \bar{\epsilon}^{\text{cs}} + 2\bar{\epsilon}^{\text{gen}} + 2\sqrt{\frac{\log(K)}{T}}$$

$$\le \inf_{h,r} \sum_k \bar{\beta}_k \cdot e_k(h, r) + c \cdot \mathbb{P}(r(x) = 1) + \bar{\epsilon}^{\text{cs}} + 2\bar{\epsilon}^{\text{gen}} + 2\sqrt{\frac{\log(K)}{T}} \quad (19)$$

$$= \inf_{h,r} \max_{\beta \in \Delta_K} \sum_k \beta_k \cdot e_k(h,r) + c \cdot \mathbb{P}(r(x)=1) + \bar{\epsilon}^{\mathrm{cs}} + 2\bar{\epsilon}^{\mathrm{gen}} + 2\sqrt{\frac{\log(K)}{T}}$$

$$\leq \inf_{h,r} \max_k e_k(h,r) + c \cdot \mathbb{P}(r(x)=1) + \bar{\epsilon}^{\mathrm{cs}} + 2\bar{\epsilon}^{\mathrm{gen}} + 2\sqrt{\frac{\log(K)}{T}}$$

$$= \inf_{h,r} R^{\mathrm{rej}}_{\mathrm{wst}}(h,r) + \bar{\epsilon}^{\mathrm{cs}} + 2\bar{\epsilon}^{\mathrm{gen}} + 2\sqrt{\frac{\log(K)}{T}}, \tag{20}$$

where in Eq. (18), we apply the bound on the excess cost-sensitive risk in line 4 of the algorithm, and in Eq. (19), we denote $\bar{\beta}_k = \frac{1}{T}\sum_{t=1}^T \beta_k^{(t)}$. We can re-write the LHS as:

$$\max_{\beta \in \Delta_K} \frac{1}{T} \sum_{t=1}^T \sum_k \beta_k \cdot \left( e_k(h^{(t)}, r^{(t)}) + c \cdot \mathbb{P}(r^{(t)}=1) \right)$$

$$= \max_{\beta \in \Delta_K} \sum_k \beta_k \cdot \mathbb{E}_{t \sim U(1,\ldots,T)} \left[ e_k(h^{(t)}, r^{(t)}) + c \cdot \mathbb{P}(r^{(t)}=1) \right]$$

$$= \max_{k \in [K]} \mathbb{E}_{t \sim U(1,\ldots,T)} \left[ e_k(h^{(t)}, r^{(t)}) + c \cdot \mathbb{P}(r^{(t)}=1) \right]$$

$$= \max_{k \in [K]} \mathbb{E}_{t \sim U(1,\ldots,T)} \left[ e_k(h^{(t)}, r^{(t)}) \right] + c \cdot \mathbb{E}_{t \sim U(1,\ldots,T)} \left[ \mathbb{P}(r^{(t)}=1) \right]$$

Substituting back into Eq. (20) completes the proof. □

## B  EQUIVALENCE BETWEEN SELECTIVE CLASSIFICATION AND L2R

Selective classification seeks to learn a classifier and rejector that minimizes the misclassification error with a constraint on the proportion of abstentions:

$$\min_{h,r} \mathbb{P}(y \neq h(x),\, r(x)=0) \quad \text{s.t.} \quad \mathbb{P}(r(x)=1) \leq b, \tag{21}$$

where $b > 0$ is an abstention budget.

**Theorem 6.** *Suppose $\mathbb{P}(y \mid x)$ and $\mathbb{P}(x)$ are continuous in $x$. For any such abstention budget $b > 0$ in the constrained problem in Eq. (21), there exists a cost $c \geq 0$ cost such that the minimizer of the L2R risk below is also a minimizer for Eq. (21):*

$$\mathbb{P}(y \neq h(x),\, r(x)=0) + c \cdot \mathbb{P}(r(x)=1).$$

The Lagrangian for the constrained problem in Eq. (21) is given by:

$$L(h,r,\lambda) = \mathbb{P}(y \neq h(x),\, r(x)=0) + \lambda \cdot (\mathbb{P}(r(x)=1) - b),$$

where $\lambda \geq 0$ is the Lagrange multiplier.

We will find it useful to state the following lemma:

**Lemma 7.** *Let $(h_\lambda^*, r_\lambda^*)$ be the minimizer of $L(h,r,\lambda)$ for Lagrange multiplier $\lambda \geq 0$. Then:*

$$\mathbb{P}(y \neq h_\lambda^*(x),\, r_\lambda^*(x)=0) \leq \mathbb{P}(y \neq h(x),\, r(x)=0),$$

*for all $(h,r)$ such that $\mathbb{P}(r(x)=1) \leq \mathbb{P}(r_\lambda^*(x)=1)$.*

*Proof.* Since $(h_\lambda^*, r_\lambda^*)$ minimizes the Lagrangian, for any $(h,r)$, $L(h_\lambda^*, r_\lambda^*, \lambda) \leq L(h,r,\lambda)$, i.e.,

$$\mathbb{P}(y \neq h_\lambda^*(x),\, r_\lambda^*(x)=0) \leq \mathbb{P}(y \neq h(x),\, r(x)=0) + \lambda \cdot (\mathbb{P}(r(x)=1) - \mathbb{P}(r_\lambda^*(x)=1)).$$

Since $\lambda \geq 0$, for any $(h,r)$ such that $\mathbb{P}(r(x)=1) \leq \mathbb{P}(r_\lambda^*(x)=1)$,

$$\mathbb{P}(y \neq h_\lambda^*(x),\, r_\lambda^*(x)=0) \leq \mathbb{P}(y \neq h(x),\, r(x)=0),$$

as desired. □

We are now ready to prove Theorem 6.

*Proof of Theorem 6.* For a fixed $\lambda \geq 0$, the following is a minimizer of the Lagrangian $L(h, r, \lambda)$:

$$h_\lambda^*(x) \in \arg\max_{y \in [L]} \mathbb{P}(y \mid x);$$

$$r_\lambda^*(x) = 1 \iff \max_{y \in [L]} \mathbb{P}(y \mid x) < 1 - \lambda.$$

The abstention rate for $r_\lambda^*(x)$ can then be written as:

$$\mathbb{P}(r_\lambda^*(x) = 1) = \int_{\max_{y \in [L]} \mathbb{P}(y|x) < 1 - \lambda} \mathbb{P}(x) \, dx.$$

Since $\mathbb{P}(y \mid x)$ and $\mathbb{P}(x)$ are continuous in $x$, we can always find a $\lambda \geq 0$ for which $\mathbb{P}(r_\lambda^*(x) = 1) = b$. Applying Lemma 7 with this choice of $\lambda$, we then have that

$$\mathbb{P}\left(y \neq h_\lambda^*(x), \, r_\lambda^*(x) = 0\right) \leq \mathbb{P}\left(y \neq h(x), \, r(x) = 0\right),$$

for all $(h, r)$ such that $\mathbb{P}(r(x) = 1) \leq b$. Setting $c = \lambda$ completes the proof. □

## C  CONDITIONAL LEARNING TO REJECT

Geifman & El-Yaniv (2017) minimize a variant of the L2R risk in Eq. (1), where the misclassification error is conditioned on samples that are not rejected:

$$R_{\mathrm{cond}}^{\mathrm{rej}}(h, r) = \mathbb{P}\left(y \neq h(x) \mid r(x) = 0\right) + c \cdot \mathbb{P}\left(r(x) = 1\right). \tag{22}$$

We show below that the optimal classifier and rejector for this *conditional* L2R risk continues to have the same form as Eq. (2), but uses a different (distribution-dependent) rejection threshold:

**Lemma 8.** *Under Assumption 1, the Bayes-optimal solution to Eq. (22) is given by:*

$$h^*(x) = \underset{y \in [L]}{\mathrm{argmax}}\, \eta_y(x); \qquad r^*(x) = 1 \iff \max_{y \in [L]} \eta_y(x) < 1 - c', \tag{23}$$

*for some distribution-dependent threshold $c' \in [0, 1]$.*

*Proof.* The conditional L2R risk in Eq. (22) can be expressed as:

$$R_{\mathrm{cond}}^{\mathrm{rej}}(h, r) = \frac{\mathbb{P}\left(y \neq h(x), r(x) = 0\right)}{1 - \mathbb{P}\left(r(x) = 1\right)} + c \cdot \mathbb{P}\left(r(x) = 1\right).$$

Minimizing this risk can be posed as an equivalent constrained optimization problem:

$$\min_{h, r, \alpha \in (0,1]} \frac{1}{\alpha} \cdot \mathbb{P}\left(y \neq h(x), r(x) = 0\right) + c \cdot \mathbb{P}\left(r(x) = 1\right) \quad \text{s.t.} \quad \mathbb{P}\left(r(x) = 1\right) = 1 - \alpha.$$

In particular, there exists an $\alpha^* \in (0, 1]$, such that the minimizer of the following constrained problem also minimizes Eq. (22):

$$\min_{h, r} \frac{1}{\alpha^*} \cdot \mathbb{P}\left(y \neq h(x), r(x) = 0\right) + c \cdot \mathbb{P}\left(r(x) = 1\right) \quad \text{s.t.} \quad \mathbb{P}\left(r(x) = 1\right) = 1 - \alpha^* \tag{24}$$

Using arguments similar to the Neyman-Pearson lemma (Neyman & Pearson, 1933), one can show that, under Assumption 1, there exists a Lagrange multiplier $\mu^* \in \mathbb{R}$ such that the minimizer to the following problem is also a minimizer to Eq. (24):

$$\min_{h, r} \frac{1}{\alpha^*} \cdot \mathbb{P}\left(y \neq h(x), r(x) = 0\right) + (c + \mu^*) \cdot \mathbb{P}\left(r(x) = 1\right),$$

or equivalently:

$$\min_{h, r} \mathbb{E}_x \left[ \frac{1}{\alpha^*} \cdot (1 - \eta_{h(x)}(x)) \cdot \mathbf{1}\left(r(x) = 0\right) + (c + \mu^*) \cdot \mathbf{1}\left(r(x) = 1\right) \right].$$

The following are then minimizers of the above objective:

$$h^*(x) \in \arg\max_{y \in [L]} \eta_y(x)$$

and

$$r^*(x) = 1 \iff \max_{y \in [L]} \eta_y(x) < 1 - \alpha^* \cdot (c + \mu^*).$$

Setting $c' = \alpha^* \cdot (c + \mu^*)$ completes the proof. □

## D  BAYES-OPTIMAL CLASSIFIERS AND REJECTORS

We derive the Bayes-optimal classifier and rejector for the balanced L2R risk (6) and the generalized L2R risk (14). We will will first consider a general formulation where we allow stochastic classifier-rejector mechanisms and then specialize to the case of deterministic classifiers and rejectors.

### D.1  CONFUSION STATISTICS

We begin with some useful definitions. Suppose $\mathcal{Y} = [n] = \{1, \ldots, n\}$. We define the confusion matrix for a classifier-rejector pair $(h, r)$ by:

$$C_{ij}(h, r) = \mathbb{E}_{(x,y)\sim\mathbb{P}}\left[\mathbf{1}\left(y = i, h(x) = j, r(x) = 0\right)\right]$$

We allow for stochastic mechanisms $F$ that are defined by a distribution over classifier-rejector pairs. We adopt the convention in Narasimhan et al. (2015a) and define the confusion matrix for a stochastic mechanism $F$ by:

$$C_{ij}(F) = \mathbb{E}_{(h,r)\sim F}\left[C_{ij}(h, r)\right].$$

We will also find it useful to define the space of all confusion matrices realizable by distributions over classifiers-rejector pairs:

$$\mathcal{C} = \left\{C(F) \in [0, 1]^{n \times n} \mid \forall \text{ distributions } F \text{ over } (h, r)\right\}.$$

It is easy to show that the set $\mathcal{C}$ is convex.

**Lemma 9.** $\mathcal{C}$ *is a convex set.*

*Proof.* For any $\mathbf{C}^{(1)}, \mathbf{C}^{(2)} \in \mathcal{C}$ and $\gamma \in [0, 1]$, we will show $\gamma \cdot \mathbf{C}^{(1)} + (1 - \gamma) \cdot \mathbf{C}^{(2)} \in \mathcal{C}$. Clearly, there exists stochastic mechanisms $F_1, F_2$ such that $\mathbf{C}^{(1)} = C(F_1)$ and $\mathbf{C}^{(2)} = C(F_2)$. Since $F' = \gamma \cdot F_1 + (1-\gamma) \cdot F_2$ represents a valid stochastic mechanism, $C(F') = \gamma \cdot \mathbf{C}^{(1)} + (1-\gamma) \cdot \mathbf{C}^{(2)} \in \mathcal{C}$. ☐

### D.2  OPTIMAL SOLUTIONS TO LINEAR MINIMIZATION OVER $\mathcal{C}$

We first state a couple of helper lemmas that characterize the optimal solutions to linear minimization problems over the set $\mathcal{C}$. Suppose we wish to minimize the following objective over stochastic mechanisms $F$:

$$\min_{F} \sum_{ij} W_{ij} \cdot C_{ij}[F],$$

for coefficients $\mathbf{W} \in \mathbb{R}^{L \times L}$. One may equivalently formulate this as a linear minimization over $\mathcal{C}$:

$$\min_{\mathbf{C} \in \mathcal{C}} \sum_{ij} W_{ij} \cdot C_{ij}. \tag{25}$$

**Lemma 10.** *Suppose Assumption 1 holds. Let* $\mathbf{W} \in \mathbb{R}^{L \times L}$ *be a coefficient matrix such that no two columns are identical. Then Eq. (25) admits a* unique *solution. Furthermore, the minimizing confusion matrix is achieved by the following deterministic classifier-rejector pair:*

$$h^*(x) \in \arg\min_{y \in [L]} \sum_{i} W_{i,y} \cdot \eta_i(x);$$

$$r^*(x) = 1 \iff \min_{y \in [L]} \sum_{i} W_{i,y} \cdot \eta_i(x) > 0.$$

*Proof.* The first part (uniqueness claim) directly follows from Lemma 24 in Narasimhan et al. (2022b). For the second part, we re-write Eq. (25) into an minimization problem over stochastic mechanisms $F$:

$$\min_{\mathbf{C} \in \mathcal{C}} \sum_{i,j} W_{ij} \cdot C_{ij} = \min_{F} \sum_{i,j} W_{ij} \cdot C_{ij}(F)$$

$$= \min_{F} \mathbb{E}_{(h,r)\sim F}\left[\sum_{i,j} W_{ij} \cdot C_{ij}(h, r)\right]$$

$$= \min_{(h,r)} \sum_{i,j} W_{ij} \cdot C_{ij}(h,r)$$

$$= \min_{(h,r)} \mathbb{E}_x \Big[ \sum_{i,j} W_{ij} \cdot \mathbb{P}(y = i, h(x) = j, r(x) = 0) \Big]$$

$$= \min_{(h,r)} \mathbb{E}_x \Big[ \sum_{i,j} W_{ij} \cdot \eta_i(x) \cdot \mathbf{1}(h(x) = j, r(x) = 0) \Big], \tag{26}$$

where the third step follows from the fact that for any function $\varphi$, the expectation $\mathbb{E}_{(h,r) \sim F} [\varphi(h,r)]$ is minimized by a point mass on $\arg \min_{(h,r)} \varphi(h,r)$.

The minimizer of Eq. (26) minimizes the term within the expectation pointwise: for any given $x$,

$$(h^*(x), r^*(x)) \in \arg\min_{y \in [L], z \in \{0,1\}} \sum_i W_{iy} \cdot \eta_i(x) \cdot \mathbf{1}(z = 0),$$

which takes the form in the statement of the lemma. $\square$

For our next helper lemma, we consider a constrained minimization problem over stochastic mechanisms $F$:

$$\min_F \sum_{i,j} W_{ij} \cdot C_{ij}[F] \quad \text{s.t.} \quad \sum_{i,j} w_i^{(k)} \cdot C_{ij}[F] = b_k, \ \forall k \in [K],$$

for coefficients $\mathbf{W} \in \mathbb{R}^{L \times L}$ and $w^{(1)}, \ldots, w^{(K)} \in \mathbb{R}^L$. Once again, we may formulate this as an equivalent constrained linear minimization over $\mathcal{C}$:

$$\min_{\mathbf{C} \in \mathcal{C}} \sum_{i,j} W_{ij} \cdot C_{ij} \quad \text{s.t.} \quad \sum_{i,j} w_i^{(k)} \cdot C_{ij} = b_k, \ \forall k \in [K]. \tag{27}$$

**Lemma 11.** *Suppose Assumption 1 holds. Let $\mathbf{W} \in \mathbb{R}^{L \times L}$ be a coefficient matrix such that no two columns are identical. Let $w^{(1)}, \ldots, w^{(K)} \in \mathbb{R}^L$ be constraint coefficients so that Eq. (27) admits a feasible solution. Then there exists multipliers $\mu^* \in \mathbb{R}^K$ such that the minimizer to Eq. (27) coincides with the minimizer to the following unconstrained linear minimization over $\mathcal{C}$:*

$$\min_{\mathbf{C} \in \mathcal{C}} \sum_{i,j} \Big( W_{ij} - \sum_{k \in [K]} \mu_k^* \cdot w_i^{(k)} \Big) \cdot C_{ij}. \tag{28}$$

*Proof.* We first write the Lagrangian for Eq. (27):

$$\mathcal{L}(\mathbf{C}, \mu) = \sum_{i,j} W_{ij} \cdot C_{ij} - \sum_{k \in [K]} \mu_k \cdot \Big( \sum_{i,j} w_i^{(k)} \cdot C_{ij} - b_k \Big), \tag{29}$$

where $\mu_1, \ldots, \mu_K \in \mathbb{R}$ are multipliers associated with the $K$ constraints.

Since the objective and constraints in Eq. (27) are linear in $\mathbf{C}$, and $\mathcal{C}$ is convex from Lemma 9, Eq. (27) is a convex minimization problem. Furthermore, since it has at least one feasible solution, Slater's condition is satisfied, and strong duality holds. As a result, there exists an optimal primal solution $\mathbf{C}^*$ that satisfies the constraints in Eq. (27), and optimal dual solution $\mu^*$ such that:

$$\mathcal{L}(\mathbf{C}^*, \mu^*) = \min_{\mathbf{C} \in \mathcal{C}} \mathcal{L}(\mathbf{C}, \mu^*) = \max_{\mu \in \mathbb{R}^K} \mathcal{L}(\mathbf{C}^*, \mu) = \sum_{i,j} W_{ij} \cdot C_{ij}^*. \tag{30}$$

Furthermore, we have that:

$$\min_{\mathbf{C} \in \mathcal{C}} \mathcal{L}(\mathbf{C}, \mu^*) = \min_{\mathbf{C} \in \mathcal{C}} \sum_{i,j} \Big( W_{ij} - \sum_{k \in [K]} \mu_k^* \cdot w_i^{(k)} \Big) \cdot C_{ij} + \omega^* = \min_{\mathbf{C} \in \mathcal{C}} \sum_{i,j} L_{ij} \cdot C_{ij} + \omega^*,$$

where $L_{ij} = W_{ij} - \sum_{k \in [K]} \mu_k^* \cdot w_i^{(k)}$ and $\omega^* = \sum_{k \in [K]} \mu_k^* \cdot b_k$ is independent of $\mathbf{C}$.

Observe that the coefficient matrix $\mathbf{L}$ on the right-hand side has no two columns identical. To see this, each column of $\mathbf{L}$ is of the form $\mathbf{L}_{:,j} = \mathbf{W}_{:,j} - \sum_{k \in [K]} \mu_k^* \cdot \mathbf{w}^{(k)}$, where $\mathbf{w}^{(k)} = [w_1^{(k)}, \ldots, w_L^{(k)}]^\top$. Since $\mathbf{W}$ has no two columns identical, i.e., $\mathbf{W}_{:j} \neq \mathbf{W}_{:j'}, \forall j \neq j'$, we have that $\mathbf{L}_{:,j} \neq \mathbf{L}_{:,j'}$.

Using Assumption 1 and the above observation, we can apply the first part of Lemma 10 to show that $\min_{\mathbf{C} \in \mathcal{C}} \mathcal{L}(\mathbf{C}, \mu^*)$ has a unique solution. Since $\mathcal{L}(\mathbf{C}^*, \mu^*) = \min_{\mathbf{C} \in \mathcal{C}} \mathcal{L}(\mathbf{C}, \mu^*)$ from Eq. (30), we have that the optimal primal solution $\mathbf{C}^*$ is a unique minimizer to $\min_{\mathbf{C} \in \mathcal{C}} \mathcal{L}(\mathbf{C}, \mu^*)$. In other words, $\mathbf{C}^*$ is the unique minimizer to the unconstrained linear problem in Eq. (28). $\square$

### D.3 BAYES-OPTIMAL CLASSIFIER AND REJECTOR FOR COST-SENSITIVE L2R

In this section, we derive the Bayes-optimal stochastic mechanism that minimizes the following cost-sensitive objective, with a penalty of $c > 0$ for abstention. Specifically, define:

$$R_{\text{cs}}^{\text{rej}}(h, r) = \sum_{k \in [K]} \beta_k \cdot \mathbb{P}\Big(y \neq h(x) \;\Big|\; r(x) = 0, \, y \in G_k\Big) + c \cdot \mathbb{P}\left(r(x) = 1\right), \qquad (31)$$

where $\beta \in \Delta_K$ are the costs associated with the $K$ groups. One can re-define Eq. (31) in terms of the confusion matrices $C(h, r)$ as follows:

$$R_{\text{cs}}^{\text{rej}}(h, r) = \sum_{k \in [K]} \beta_k \cdot \frac{\mathbb{P}\big(y \neq h(x), \, r(x) = 0, \, y \in G_k\big)}{\mathbb{P}\big(r(x) = 0, \, y \in G_k\big)} + c \cdot \mathbb{P}\left(r(x) = 1\right)$$

$$= \sum_{k \in [K]} \beta_k \cdot \frac{\sum_{i \in G_k} \sum_{j \neq i} C_{ij}(h, r)}{\sum_{i \in G_k} \sum_{j \in [L]} C_{ij}(h, r)} + c \cdot \Big(1 - \sum_{i,j} C_{ij}(h, r)\Big).$$

For a stochastic mechanism $F$, we can similarly define

$$R_{\text{cs}}^{\text{rej}}(F) = \sum_{k \in [K]} \beta_k \cdot \frac{\sum_{i \in G_k} \sum_{j \neq i} C_{ij}(F)}{\sum_{i \in G_k} \sum_{j \in [L]} C_{ij}(F)} + c \cdot \Big(1 - \sum_{i,j} C_{ij}(F)\Big). \qquad (32)$$

We derive an optimal stochastic classifier that minimizes Eq. (32):

**Theorem 12.** *Under Assumptions 1 and 2, there exists group-specific parameters $\alpha^* \in (0, 1)^K$ and $\mu^* \in \mathbb{R}^K$ such that the following deterministic classifier-rejector pair is an optimal solution to Eq. (32):*

$$h^*(x) = \arg\max_{y \in [L]} \frac{\beta_{[y]}}{\alpha_{[y]}^*} \cdot \eta_y(x);$$

$$r^*(x) = 1 \iff \max_{y \in [L]} \frac{\beta_{[y]}}{\alpha_{[y]}^*} \cdot \eta_y(x) < \sum_{y' \in [L]} \left(\frac{\beta_{[y']}}{\alpha_{[y']}^*} - \mu_{[y']}^*\right) \cdot \eta_{y'}(x) - c, \qquad (33)$$

*where $[y]$ is the index of the group class $y$ belongs to. Furthermore, $\alpha_k^* = \mathbb{P}\left(r^*(x) = 0, \, y \in G_k\right)$.*

*Proof.* From the boundedness condition in Assumption 2, we know that there exists $\alpha^* \in (0, 1)^K$ such that the minimizer to the following problem also minimizes Eq. (32):

$$\min_F \sum_{k \in [K]} \frac{\beta_k}{\alpha_k^*} \cdot \sum_{i \in G_k} \sum_{j \neq i} C_{ij}(F) + c \cdot \Big(1 - \sum_{i,j} C_{ij}(F)\Big) \qquad (34)$$

$$\text{s.t.} \quad \sum_{i \in G_k} \sum_{j \in [L]} C_{ij}(F) = \alpha_k^*, \quad \forall k \in [K].$$

One may equivalently formulate a constrained linear minimization problem over $\mathcal{C}$:

$$\min_{\mathbf{C} \in \mathcal{C}} \sum_{k \in [K]} \frac{\beta_k}{\alpha_k^*} \cdot \sum_{i \in G_k} \sum_{j \neq i} C_{ij} + c \cdot \Big(1 - \sum_{i,j} C_{ij}\Big) \qquad (35)$$

$$\text{s.t.} \quad \sum_{i \in G_k} \sum_{j \in [L]} C_{ij} = \alpha_k^*, \quad \forall k \in [K].$$

One can now directly apply the result in Lemma 11 with $W_{ij} = \sum_{k \in [K]} \frac{\beta_k}{\alpha_k^*} \cdot \mathbf{1}(i \in G_k, j \neq i) - c$ and $w_i^{(k)} = \mathbf{1}(i \in G_k), \forall i \in [L]$, noting that no two columns of $\mathbf{W}$ are identical. Consequently, we have that there exists multipliers $\mu^* \in \mathbb{R}^K$ such that the minimizer to the following unconstrained linear minimization over $\mathcal{C}$ also minimizes Eq. (35):

$$\min_{\mathbf{C} \in \mathcal{C}} \sum_{i,j} \Big(W_{ij} - \sum_{k \in [K]} \mu_k^* \cdot w_i^{(k)}\Big) \cdot C_{ij}.$$

Applying Lemma 10, we have that the above problem admits a unique minimizer $\mathbf{C}^*$ which is achieved by the following deterministic classifier-rejector pair:

$$h^*(x) \in \arg\min_{y \in [L]} \sum_i \left( W_{iy} - \sum_{k \in [K]} \mu_k^* \cdot w_i^{(k)} \right) \cdot \eta_i(x);$$

$$r^*(x) = 1 \iff \min_{y \in [L]} \sum_i \left( W_{iy} - \sum_{k \in [K]} \mu_k^* \cdot w_i^{(k)} \right) \cdot \eta_i(x) > 0.$$

Substituting for $\mathbf{W}$ and $\mathbf{w}_i^{(k)}$ then gives us:

$$h^*(x) \in \arg\min_{y \in [L]} \sum_{k \in [K]} \sum_{i \in G_k} \left( \frac{\beta_k}{\alpha_k^*} \cdot \mathbf{1}(y \neq i) - \mu_k^* \right) \cdot \eta_i(x) - c;$$

$$r^*(x) = 1 \iff \min_{y \in [L]} \sum_{k \in [K]} \sum_{i \in G_k} \left( \frac{\beta_k}{\alpha_k^*} \cdot \mathbf{1}(y \neq i) - \mu_k^* \right) \cdot \eta_i(x) > c.$$

The optimal classifier $h^*$ then simplifies to:

$$h^*(x) \in \arg\max_{y \in [L]} \sum_{k \in [K]} \sum_{y \in G_k} \frac{\beta_k}{\alpha_k^*} \cdot \eta_y(x).$$

Since each class $y$ belongs exactly one group, this is the same as:

$$h^*(x) \in \arg\max_{y \in [L]} \frac{\beta_{[y]}}{\alpha_{[y]}^*} \cdot \eta_y(x),$$

where $[y]$ is the index of the group class $y$ belongs to.

Similarly, the optimal rejector $r^*$ simplifies to

$$r^*(x) = 1 \iff \min_{y \in [L]} \sum_{k \in [K]} \sum_{i \in G_k} \left( \frac{\beta_k}{\alpha_k^*} - \mu_k^* \right) \cdot \eta_i(x) - \sum_{k \in [K]} \sum_{y \in G_k} \frac{\beta_k}{\alpha_k^*} \cdot \eta_y(x) > c,$$

which is the same as:

$$r^*(x) = 1 \iff \max_{y \in [L]} \frac{\beta_{[y]}}{\alpha_{[y]}^*} \cdot \eta_y(x) < \sum_{y' \in [L]} \left( \frac{\beta_{[y']}}{\alpha_{[y']}^*} - \mu_{[y']}^* \right) \cdot \eta_{y'}(x) - c,$$

as desired. $\qquad\square$

### D.4 SPECIAL CASE OF BINARY GROUPS WITH BINARY LABELS

In the special case where we have binary labels, each in a separate group, the rejector in Theorem 12 takes a much simpler form, with a constant threshold for each class.

**Corollary 13.** *Suppose $\mathcal{Y} = \mathcal{G} = \{0, 1\}$. Under Assumption 1, the optimal classifier and rejector for Eq. (32) simplifies to applying class-specific thresholds $\tau_0^*, \tau_1^* \in \mathbb{R}$ and $\gamma^* \in (0, 1)$:*

$$h^*(x) = \mathbf{1}\left( \eta_1(x) > \gamma^* \right); \quad r^*(x) = 1 \iff \eta_{h^*(x)}(x) < \tau_{h^*(x)}^*.$$

*Proof.* When $\mathcal{Y} = \mathcal{G} = \{0, 1\}$, we have from Theorem 12 that the optimal classifier takes the form:

$$h^*(x) = \arg\max\left\{ \frac{\beta_0}{\alpha_0^*} \cdot \eta_0(x), \frac{\beta_1}{\alpha_1^*} \cdot \eta_1(x) \right\} = \mathbf{1}\left( \frac{\beta_0}{\alpha_0^*} \cdot \eta_0(x) < \frac{\beta_1}{\alpha_1^*} \cdot \eta_1(x) \right) = \mathbf{1}\left( \eta_1(x) > \gamma^* \right),$$

where $\gamma^* = \frac{\beta_0/\alpha_0^*}{\beta_0/\alpha_0^* + \beta_1/\alpha_1^*}$, as desired; the last equality uses the fact that $\eta_0(x) + \eta_1(x) = 1$.

Similarly, the optimal rejector is given by:

$$r^*(x) = 1 \iff \frac{\beta_{h^*(x)}}{\alpha_{h^*(x)}^*} \cdot \eta_{h^*(x)}(x) < \left( \frac{\beta_0}{\alpha_0^*} - \mu_0^* \right) \cdot \eta_0(x) + \left( \frac{\beta_1}{\alpha_1^*} - \mu_1^* \right) \cdot \eta_1(x) - c. \quad (36)$$

Consider the case when $h^*(x) = 1$. We can then use the fact that $\eta_0(x) + \eta_1(x) = 1$ to re-write the right-hand side of Eq. (36) as:

$$\frac{\beta_1}{\alpha_1^*} \cdot \eta_1(x) < \left(\frac{\beta_0}{\alpha_0^*} - \mu_0^*\right) \cdot (1 - \eta_1(x)) + \left(\frac{\beta_1}{\alpha_1^*} - \mu_1^*\right) \cdot \eta_1(x) - c,$$

which simplifies to:

$$\eta_1(x) < \tau_1^*,$$

where

$$\tau_1^* = \frac{\frac{\beta_0}{\alpha_0^*} - \mu_0^* - c}{\frac{\beta_0}{\alpha_0^*} + \mu_1^* - \mu_0^*}.$$

Similarly, when $h^*(x) = 1$, the right-hand side of Eq. (36) simplifies to:

$$\eta_0(x) < \tau_0^*,$$

where

$$\tau_0^* = \frac{\frac{\beta_1}{\alpha_1^*} - \mu_1^* - c}{\frac{\beta_1}{\alpha_1^*} + \mu_0^* - \mu_1^*},$$

which completes the proof. $\qquad\square$

### D.5 BAYES-OPTIMAL CLASSIFIER AND REJECTOR FOR GENERALIZED L2R

We next derive the Bayes-optimal classifier and rejector for the generalized L2R risk in Eq. (14), which we restate below:

$$R_{\text{gen}}^{\text{rej}}(h, r) = \psi\left(e_1(h, r), \ldots, e_K(h, r)\right) + c \cdot \mathbb{P}\left(r(x) = 1\right), \tag{37}$$

where $\psi : [0, 1]^K \to \mathbb{R}_+$ is a general function, and encompasses several common metrics (Lipton et al., 2014; Menon et al., 2013; Narasimhan et al., 2015a).

One may equivalently write Eq. (37) in terms of the confusion matrix $C(h, r)$:

$$R_{\text{gen}}^{\text{rej}}(h, r) = \psi\left(\frac{\sum_{i \in G_1} \sum_{j \neq i} C_{ij}(h, r)}{\sum_{i \in G_1} \sum_{j \in [L]} C_{ij}(h, r)}, \ldots, \frac{\sum_{i \in G_K} \sum_{j \neq i} C_{ij}(h, r)}{\sum_{i \in G_K} \sum_{j \in [L]} C_{ij}(h, r)}\right) + c \cdot \left(1 - \sum_{i,j} C_{ij}(h, r)\right).$$

Allowing for stochastic mechanisms $F$ that are defined by a distribution over classifier-rejector pairs, we would like to minimize:

$$\min_F \psi\left(\frac{\sum_{i \in G_1} \sum_{j \neq i} C_{ij}(F)}{\sum_{i \in G_1} \sum_{j \in [L]} C_{ij}(F)}, \ldots, \frac{\sum_{i \in G_K} \sum_{j \neq i} C_{ij}(F)}{\sum_{i \in G_K} \sum_{j \in [L]} C_{ij}(F)}\right) + c \cdot \left(1 - \sum_{i,j} C_{ij}(F)\right),$$

or equivalently:

$$\min_F \psi\left(\frac{\phi_1(C(F))}{\pi_1(C(F))}, \ldots, \frac{\phi_K(C(F))}{\pi_K(C(F))}\right) + c \cdot \left(1 - \sum_k \pi_k(C(F))\right), \tag{38}$$

where $\phi_k(\mathbf{C}) = \sum_{i \in G_k} \sum_{j \neq i} C_{ij}(F)$ and $\pi_k(\mathbf{C}) = \sum_{i \in G_k} \sum_{j \in [L]} C_{ij}(F)$.

**Theorem 7** (restated). *For any non-linear $\psi$, there exists coefficients $u^{(i)}, v^{(i)} \in \mathbb{R}^K$, $i \in [K+1]$ and a parameter $\nu \in \Delta_{K+1}$, such that, the generalized L2R in Eq. (14) is minimized by a stochastic mechanism that for any $x$, predicts using $(h^{(i)}, r^{(i)})$ with probability $\nu_i$:*

$$h^{(i)}(x) \in \arg\max_y u_y^{(i)} \cdot \eta_y(x), \ \ i \in [K+1];$$

$$r^{(i)}(x) = 1 \iff \max_y u_y^{(i)} \cdot \eta_y(x) < \sum_{y'} v_{y'}^{(i)} \cdot \eta_{y'}(x) - c, \ \ i \in [K+1].$$

*Proof.* One may reformulate Eq. (38) as an optimization over the space of all confusion statistics realizable through distribution over classifiers-rejector pairs $\mathcal{C}$, or equivalently over a smaller space of transformed confusion matrices:

$$\mathcal{C}' = \{(\boldsymbol{\phi}, \boldsymbol{\pi}), \text{ where } \boldsymbol{\phi} = (\phi_1(\mathbf{C}), \dots, \phi_K(\mathbf{C})), \text{ and } \boldsymbol{\pi} = (\pi_1(\mathbf{C}), \dots, \pi_K(\mathbf{C})) \,|\, \forall \mathbf{C} \in \mathcal{C}\}.$$

We can then solve Eq. (38) by solving:

$$\min_{(\boldsymbol{\phi}, \boldsymbol{\pi}) \in \mathcal{C}'} \psi\left(\frac{\phi_1}{\pi_1}, \dots, \frac{\phi_K}{\pi_K}\right) + c \cdot \left(1 - \sum_k \pi_k\right). \tag{39}$$

We know from Lemma 9 that $\mathcal{C}$ is convex; consequently, it is easy to see that $\mathcal{C}'$ is also convex. Therefore any point $(\boldsymbol{\phi}, \boldsymbol{\pi}) \in \mathcal{C}'$ can be expressed as a convex combination of extreme points of $\mathcal{C}'$. In fact, following Proposition 10 in Narasimhan et al. (2022b), any $(\boldsymbol{\phi}, \boldsymbol{\pi}) \in \mathcal{C}'$ can be expressed as convex combination of $K + 1$ extreme points $(\boldsymbol{\phi}^{(1)}, \boldsymbol{\pi}^{(1)}), \dots (\boldsymbol{\phi}^{(K+1)}, \boldsymbol{\pi}^{(K+1)})$, each of which is a unique linear minimizer over $\mathcal{C}'$:

$$(\boldsymbol{\phi}^{(i)}, \boldsymbol{\pi}^{(i)}) \in \arg\min_{(\boldsymbol{\phi}, \boldsymbol{\pi}) \in \mathcal{C}'} \sum_k w_k^{(i)} \cdot \phi_k + \sum_k q_k^{(i)} \cdot \pi_k$$

for coefficients $w^{(i)}, q^{(i)} \in \mathbb{R}^K$, $i \in [K + 1]$. Therefore this is also true for the minimizer $(\boldsymbol{\phi}^*, \boldsymbol{\pi}^*) \in \mathcal{C}'$ of Eq. (39).

All that remains to show is that these $K + 1$ linear minimizers are realized by $K + 1$ classifier-rejector pairs of the form in the theorem statement. To show this, we note that for coefficients $w^{(i)}, q^{(i)} \in \mathbb{R}^K$, minimizing the linear objective over $\mathcal{C}'$:

$$\min_{(\boldsymbol{\phi}, \pi) \in \mathcal{C}'} \sum_k w_k^{(i)} \cdot \phi_k + \sum_k q_k^{(i)} \cdot \pi_k$$

is equivalent to minimizing the same objective over the boundary points of $\mathcal{C}'$, and equivalently, over deterministic classifier-rejector pairs:

$$\min_{h, r} \sum_k w_k^{(i)} \cdot \phi_k(C(h, r)) + \sum_k q_k^{(i)} \cdot \pi_k(C(h, r)). \tag{40}$$

Expanding the above objective, we get:

$$\sum_k w_k^{(i)} \cdot \mathbb{E}\left[\mathbf{1}(y \neq h(x), r(x) = 0, y \in G_k)\right] + \sum_k q_k^{(i)} \cdot \mathbb{E}\left[\mathbf{1}(r(x) = 0, y \in G_k)\right],$$

which is the same as:

$$\mathbb{E}_x\left[\sum_k w_k^{(i)} \sum_{y \in G_k} \eta_y(x) \cdot \mathbf{1}(y \neq h(x), r(x) = 0) + \sum_k q_k^{(i)} \sum_{y \in G_k} \eta_y(x) \cdot \mathbf{1}(r(x) = 0)\right],$$

or:

$$\mathbb{E}_x\left[\left(\sum_y w_{[y]}^{(i)} \cdot \eta_y(x) - w_{[h(x)]}^{(i)} \cdot \eta_{h(x)}(x) + \sum_y q_{[y]}^{(i)} \cdot \eta_y(x)\right) \cdot \mathbf{1}(r(x) = 0)\right].$$

The minimizer of Eq. (40) then takes the form:

$$h^{(i)}(x) \in \arg\max_{y \in [L]} w_{[y]}^{(i)} \cdot \eta_y(x);$$

$$r^{(i)}(x) = 1 \iff \max_{y \in [L]} w_{[y]}^{(i)} \cdot \eta_y(x) < \sum_{y \in [L]} (w_{[y]}^{(i)} + q_{[y]}^{(i)}) \cdot \eta_y(x).$$

Setting $u_y^{(i)} = w_{[y]}^{(i)}$ and $v_y^{(i)} = w_{[y]}^{(i)} + q_{[y]}^{(i)} + c$, the above is equivalent to:

$$h^{(i)}(x) \in \arg\max_{y \in [L]} u_y^{(i)} \cdot \eta_y(x);$$

$$r^{(i)}(x) = 1 \iff \max_{y \in [L]} u_y^{(i)} \cdot \eta_y(x) < \sum_{y \in [L]} v_y^{(i)} \cdot \eta_y(x) - c.$$

It thus follows that the minimizer $(\boldsymbol{\phi}^*, \boldsymbol{\pi}^*) = (\phi(C(F^*)), \pi(C(F^*)))$ to Eq. (39) is realized by a stochastic mechanism $F^*$ that randomizes over the $K + 1$ classifier-rejector pairs $(h^{(1)}, r^{(1)}), \dots, (h^{(K+1)}, r^{(K+1)})$. $\qquad\square$

---

**Algorithm 3** Lagrangian-based Plug-in for Balanced Error

---

1: **Input:** Rejection cost $c$, Pre-trained $p : \mathcal{X} \to \Delta_L$, Sample $S$

2: **Parameters:** Iterations $M, T$, Step-sizes $\xi_\mu, \xi_\alpha$, Initial $\alpha^{(0)} \in (0, K)^K$, Lower bound $\kappa > 0$

3: **For** $m = 0$ to $M$

4:     $h^{(m)}(x) = \arg\max_{y \in [L]} u_y^{(m)} \cdot p_y(x)$, where $u_y^{(m)} = \frac{1}{\alpha_{[y]}^{(m)}}$

5:     Initialize $\mu^{(m,0)} \in \mathbb{R}^K$

6:     **For** $t = 0$ to $T - 1$

7:         $\mu_k^{(m,t+1)} = \mu_k^{(m,t)} + \xi_\mu \cdot \left( \alpha_i^{(m)} - \frac{K}{|S|} \sum_{(x,y) \in S} \mathbf{1}\left( r(x) = 0, y \in G_k \right) \right), \; \forall k \in [K]$

8:         $r^{(m,t+1)}(x) = 1 \iff \max_{y \in [L]} u_y^{(m)} \cdot p_y(x) < \sum_{y' \in [L]} v_{y'}^{(m)} \cdot p_{y'}(x) - c,$

9:                 where $v_y^{(m)} = \frac{1}{\alpha_{[y]}^{(m)}} - \mu_{[y]}^{(m,t+1)}.$

10:     **End For**

11:     $\alpha_k^{(m+1)} = \alpha_k^{(m)} - \xi_\alpha \cdot \left( \mu_k^{(m,T)} - \frac{1}{(\alpha_k^{(m)})^2} \cdot \hat{e}_k(h^{(m)}, r^{(m,T)}) \right)$

12:     $\alpha_k^{(m+1)} = \text{proj}_{[\kappa, \, K-\kappa]}\left( \alpha_k^{(m)} \right)$, where $\text{proj}_{[a,b]}(z) = \max\{\min\{z, b\}, a\}$

13: **End For**

14: **Return:** $(h^{(M)}, r^{(M,T)})$

---

## E    PLUG-IN APPROACH: FURTHER DETAILS

We provide further details about the plug-in approach in §4.

### E.1    RE-PARAMETERIZING THE REJECTOR FOR THE BALANCED ERROR

One can further prune the search space over multipliers $\hat{\mu}$ by re-parameterizing the rejection criterion in Eq. (13) to search over only $K - 1$ parameters. To this end, using the fact that $\sum_{y \in [L]} p_y(x) = 1$, we re-write the criterion as:

$$\max_{y \in [L]} \frac{1}{\hat{\alpha}_{[y]}} \cdot p_y(x) < \sum_{y' \in [L]} \frac{1}{\hat{\alpha}_{[y']}} \cdot p_{y'}(x) - \sum_{y' \in [L]} (\hat{\mu}_{[y']} - \hat{\mu}_K) \cdot p_{y'}(x) - \hat{\mu}_K \sum_{y' \in [L]} p_{y'}(x) - c,$$

which is the same as:

$$\max_{y \in [L]} \frac{1}{\hat{\alpha}_{[y]}} \cdot p_y(x) < \sum_{y' \in [L]} \frac{1}{\hat{\alpha}_{[y']}} \cdot p_{y'}(x) - \sum_{k \in [K-1]} (\hat{\mu}_k - \hat{\mu}_K) \cdot \sum_{y' \in G_k} p_{y'}(x) - \hat{\mu}_K - c.$$

This can be equivalently re-paramterized as:

$$\max_{y \in [L]} \frac{1}{\hat{\alpha}_{[y]}} \cdot p_y(x) < \sum_{y' \in [L]} \frac{1}{\hat{\alpha}_{[y']}} \cdot p_{y'}(x) - \sum_{k \in [K-1]} \sum_{y' \in G_k} \hat{\lambda}_k \cdot p_{y'}(x) - \hat{\tau},$$

where $\hat{\lambda}_k = \hat{\mu}_k - \hat{\mu}_K$, for $k \in [K-1]$, and $\hat{\tau} = \hat{\mu}_K - c$.

Note that in practice, one is often prescribed a target rejection rate, and picks the rejection cost $c$ to achieve this rejection rate. For fixed $\hat{\lambda}_1, \ldots, \hat{\lambda}_{K-1}$, one may equivalently pick the threshold $\hat{\tau}$ (instead of cost $c$) to reach the desired rejection rate. So given a target rejection rate, all one needs to do is to apply a brute-force search to tune the remaining $K - 1$ unknowns to minimize the balanced error on a validation set, with the threshold $\hat{\tau}$ set to a percentile that matches the desired rejection rate. When $K = 2$ (head and tail), i.e., $\mathcal{G} = \{0, 1\}$, this would mean that we only have a *single* parameter $\hat{\lambda}_0 = \hat{\mu}_0 - \hat{\mu}_1$ to tune, which can be efficiently done using a simple line search.

### E.2    LAGRANGIAN-BASED PLUG-IN APPROACH

As an alternative to plug-in approach in Algorithm 1, we explore a Lagrangian-based approach that handles the constraints more intricately. We begin by introducing multipliers $\mu \in \mathbb{R}^K$ for the $K$

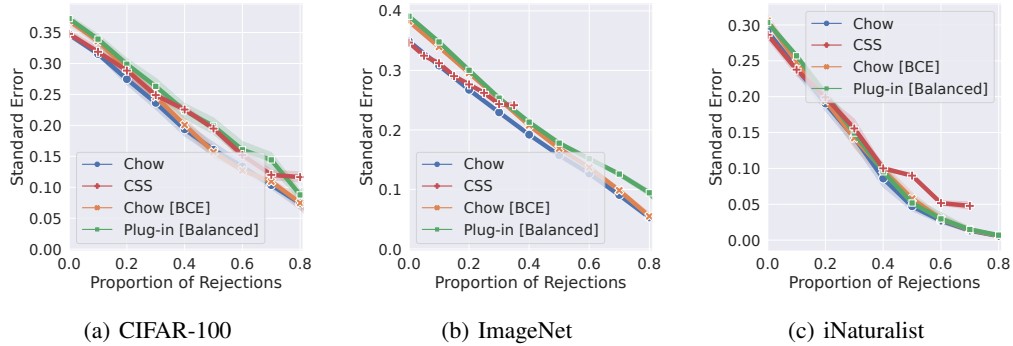

(a) CIFAR-100          (b) ImageNet          (c) iNaturalist

Figure 4: Balanced L2R: Standard 0-1 error as a function of proportion of rejections.

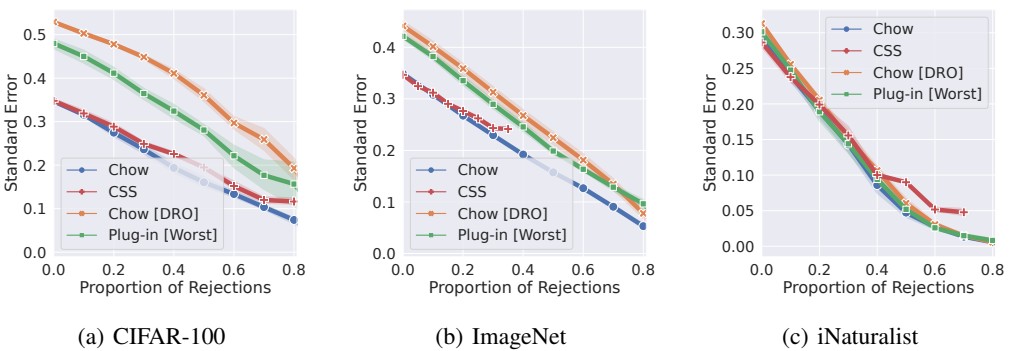

(a) CIFAR-100          (b) ImageNet          (c) iNaturalist

Figure 5: Worst-group L2R: Standard 0-1 error as a function of proportion of rejections.

constraints in the optimization problem in Eq. (7) and writing out the Lagrangian:

$$\mathcal{L}(h, r, \alpha; \mu) = \sum_{k \in [K]} \frac{1}{\alpha_k} \cdot \mathbb{P}\left(y \neq h(x), r(x) = 0, y \in G_k\right) + c \cdot \mathbb{P}\left(r(x) = 1\right)$$
$$- \sum_{k \in [K]} \mu_k \cdot \left(K \cdot \mathbb{P}\left(r(x) = 0, y \in G_k\right) - \alpha_k\right),$$

and formulate an equivalent saddle point optimization problem:

$$\min_{\alpha, h, r} \max_{\mu \in \mathbb{R}^K} \mathcal{L}(h, r, \alpha; \mu). \tag{41}$$

For a fixed $\alpha$, we can solve the min-max problem over $h, r$ and $\mu$ by alternating between gradient-ascent updates on the multipliers $\mu$:

$$\mu_k^{(t+1)} = \mu_k^{(t)} + \xi_\mu \cdot \left(\alpha_k - K \cdot \mathbb{P}\left(r(x) = 0, y \in G_k\right)\right), \ \forall k \in [K],$$

for step size $\xi_\mu > 0$, and a full minimization over $h$ and $r$. The procedure outlined in Algorithm 3 then solves the saddle point optimization problem in Eq. (41) by performing gradient-descent on $\alpha$, and the combination of the gradient-ascent and full minimization procedure described above over $h, r$ and $\mu$.

It is worth noting that this Lagrangian procedure is made possible by viewing Eq. (7) as an optimization problem over the space of the space of realizable confusion statistics $\mathcal{C}$ (Narasimhan, 2018; Narasimhan et al., 2022b), which we can then show is a continuous constrained optimization problem that is amenable be solved with Lagrangian style updates.

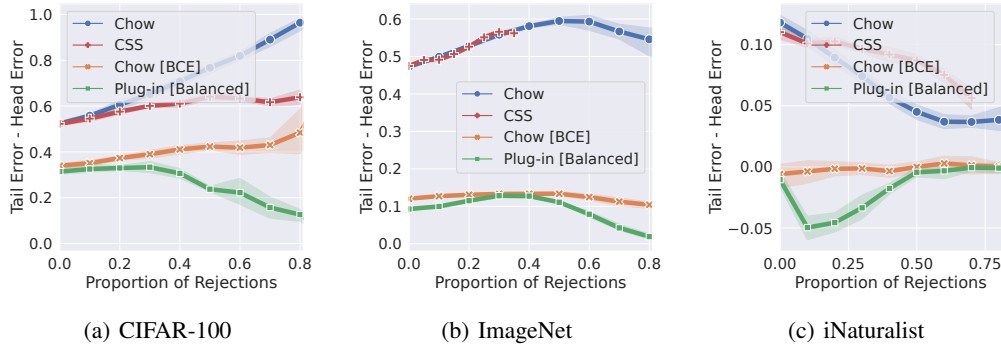

Figure 6: Balanced L2R: Difference between tail and head errors as a function of proportion of rejections.

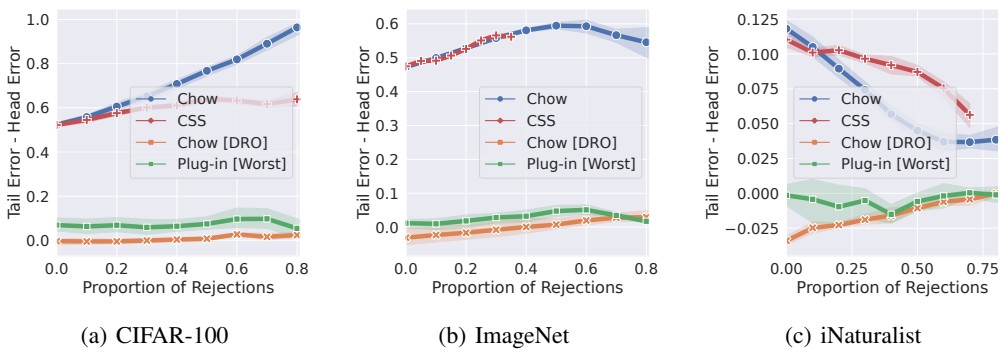

Figure 7: Worst-group L2R: Difference between tail and head errors as a function of proportion of rejections.

## F    ADDITIONAL EXPERIMENTAL RESULTS

We discuss the data splits, hyper-parameter choices and further implementation details in our experiments. We also include additional experimental plots. All results are averaged over **5 trials**. We report 95% confidence intervals in Table 2.

### F.1    HYPER-PARAMETER CHOICES

We apply the same data pre-processing as Menon et al. (2021a). We trained all the models using SGD with a momentum of 0.9 and a weight decay of $10^{-4}$. We summarise the hyper-parameter choices in Table 3 below.

| Dataset | #classes | $n_{\text{train}}$ | $n_{\text{test}}$ | Tail prop. | Model | Base LR | Schedule | Epochs | Batch size |
|---|---|---|---|---|---|---|---|---|---|
| CIFAR-100-LT | 100 | 50,000 | 10,000 | 0.03 | CIFAR ResNet-32 | 0.4 | anneal | 256 | 128 |
| ImageNet-LT | 1000 | 1,281,167 | 50,000 | 0.02 | ResNet-50 | 0.4 | cosine | 200 | 512 |
| iNaturalist 2018 | 8143 | 437,513 | 24,426 | 0.12 | ResNet-50 | 0.4 | cosine | 200 | 1024 |

Table 3: Dataset details and hyper-parameter choices.

For CIFAR-100, we apply a warm up with a linear learning rate for 15 steps until we reach the base learning rate. We apply a learning rate decay of 0.1 at the 96th, 192nd and 224th epochs.

For ImageNet and iNaturalist, we apply a warm up with a linear learning rate for 5 steps until we reach the base learning rate. We apply a learning rate decay of 0.1 at the 45th, 100th and 150th epochs.

## F.2 Dataset Splits

The train, test and validations samples all follow the same long-tailed label distributions. Each dataset comes with an original test set with equal proportions of all labels. *We re-weight the samples in the original test set to replicate the same label proportions as the training set*. Furthermore, we hold out 20% of the original test set as a validation sample and use the remaining as the test sample.

## F.3 Implementation Details

For the proposed plug-in method for the balanced error (Algorithm 1), we set the number of iterations $M$ to 10, and tune the single parameter $\hat{\lambda}_0 = \hat{\mu}_0 - \hat{\mu}_1$ (see Appendix E.1) from three values $\{1, 6, 11\}$. For the proposed plug-in method for the worst-group error (Algorithm 2), we set the number of outer iterations $T$ to 25 and the step-size $\xi$ to 1; for the inner call to Algorithm 1, we set $M$ to 10 and tune the parameter $\hat{\lambda}_0$ from $\{1, 6, 11\}$.

We implement the cost-sensitive softmax (CSS) loss of Mozannar & Sontag (2020) described in Equation 3 in their paper. We set the cost vector $c(1), \ldots, c(L + 1)$ in this loss to $c(i) = \mathbf{1}(y \neq i), \forall i \in [L]$ and $c(L + 1)$ to the rejection cost $c$.

For Chow [Balanced], there exists a plethora of options in the long-tail literature to train the base model (see §2.2 for a brief list of representative approaches). We pick the logit-adjusted cross-entropy loss of Menon et al. (2021a) as a representative approach.

For Chow [DRO], as prescribed in Jones et al. (2021), we use the group DRO approach proposed in Sagawa et al. (2020), but modify the inner weighted minimization step to use a logit-adjusted loss (Narasimhan & Menon, 2021; Wang et al., 2023). This modification was necessary to adapt the approach of Sagawa et al. (2020) to multi-class classification with a large number of classes. Their method requires tuning a step-size for DRO and a regularization parameter $C$. We pick these parameters from the sets $\{0.0005, 0.001\}$ and $\{1.0, 2.0\}$ respectively, by maximizing the worst-group error on the validation sample.

## F.4 Trade-off Plots: Further Details

To generate the trade-off plots in Figure 3–7 for each method, we train rejectors with different rejection rates. For this, one may vary the cost parameter $c$, or equivalently, vary the rejection threshold in the case of Chow's rule and the threshold $\hat{\tau}$ in the proposed plug-in method (see Appendix E.1). For a target rejection rate, this can be done with a simple *percentile* computation.

For the CSS baseline, we re-train a model for different values of $c$ in $\{0.0, 0.1, 0.5, 0.75, 0.85, 0.91, 0.95, 0.97, 0.99\}$. Unfortunately, we find that with the ImageNet and iNaturalist datasets, the resulting models do not exceed a rejection rate of 40% and 75% respectively, despite us using the entire range of $c \in [0, 1]$.

## F.5 Additional Plots

We present additional plots of the standard 0-1 error as a function of rejection rate for different methods in Figures 4–5. We also present plots of the difference between the tail and head errors as a function of rejection rates in Figures 6–7.

## F.6 Comparison to Exhaustive Search over $\alpha$

We repeated the CIFAR-100 experiments in §6 choosing the multiplier parameter from a finer grid of values: $\{0.25, 0.5, 0.75, \ldots, 11\}$. We additionally replaced our heuristic procedure in §4.2 with an exhaustive grid search over group prior $\alpha$, choosing the head-to-tail ratio from the range $\{0, 0.01, 0.02, \ldots, 1.0\}$. We pick the parameter combination that yields the lowest balanced error on the held-out validation sample.

We show below the balanced error on the test and validation samples. Although the grid search provides gains on the validation set, on the test set, our heuristic approach to picking the multipliers

and group priors yields metrics that are only slightly worse than grid-search. Moreover, the differences are within standard errors.

|  | Test | Validation |
| --- | --- | --- |
| Chow | $0.509 \pm 0.002$ | $0.498 \pm 0.011$ |
| Plug-in [Balanced] | $0.291 \pm 0.008$ | $0.282 \pm 0.007$ |
| Plug-in [Balanced, grid search] | $0.284 \pm 0.007$ | $0.264 \pm 0.008$ |

