# OpenReview forum: "Learning to Reject Meets Long-tail Learning"
_ICLR.cc/2024/Conference — ICLR 2024 spotlight_

### Official Review · Reviewer_91tX · 2023-10-25

**Soundness:** 3 good
**Presentation:** 4 excellent
**Contribution:** 3 good
**Rating:** 8
**Confidence:** 3

**Summary:**

This paper works on learning to reject (L2R) under long-tailed scenarios. They find that Chow's rule is suboptimal for this setting and propose a new approach with theoretical guarantees. Extensive experiments on benchmark datasets validate the effectiveness of the proposed approach.

**Strengths:**

- The paper is very well written. Even those unfamiliar with L2R can easily understand the setting.
- The setting is novel and meaningful, since the long-tailed setting is quite common but rarely studied in the context of L2R.
- The proposed method is novel and interesting, and the theoretical results are not trivial.

**Weaknesses:**

- If I understand correctly, the solution of the optimization problem in equation (9) seems to be heuristic. Is $\mu$ not optimized, but selected from a predefined set? I am afraid that such solutions may not lead to an optimal solution. Will the authors provide some theoretical or empirical analysis of this problem and prove that the solution can be close to optimal?

- What is the candidate value of $\mu$? The authors say that they propose to do a grid search over $\mu$, but what's the search space? If the search space is very large, the speed of the algorithm may be slow.

- In the experimental section, only two compared methods are used. I am not sure if more compared methods are needed to validate the effectiveness of the proposal.

**Questions:**

Please see the "Weaknesses" section.

---

> ### Author Response · Authors · 2023-11-22
> **Response to Reviewer 91tX**
>
> We thank the reviewer for their encouraging comments and questions.
>
> > What is the candidate value of $\mu$? The authors say that they propose to do a grid search over $\mu$, but what's the search space? If the search space is very large, the speed of the algorithm may be slow.
>
> The reviewer is correct that we apply a grid search to choose $\mu$. However, the search space we use is very small, as detailed below:
>
> - First, using the re-parametrization trick described in Appendix C.1, for $K$ groups, we only need to search over $K-1$ multiplier parameters. For our experiments, we consider settings with two groups (head and tail), requiring us to tune **only one** multiplier parameter.
>
> - Second, even for this single parameter, we find sufficient to use a **small search space**. In fact, as noted in Appendix D.3, we pick the multiplier from the small range {1, 6, 11}.
>
> - Third, recall that ours is a post-hoc approach that constructs a simple rejection rule from a **pre-trained base model**. When varying the multiplier parameter, we are merely changing the rejection rule in Equation 15 without re-training the base model. Therefore carrying out a search over multipliers can be implemented efficiently when $K$ is small.
>
> > If I understand correctly, the solution of the optimization problem in equation (9) seems to be heuristic. Is not optimized, but selected from a predefined set? I am afraid that such solutions may not lead to an optimal solution. Will the authors provide some theoretical or empirical analysis of this problem and prove that the solution can be close to optimal?
>
> To confirm that it suffices to use a small search space for the multipliers, we repeated the CIFAR-100 experiments by picking the multiplier from a fine-grained grid {0.25, 0.5, 0.75, …, 11}. We additionally replaced our heuristic procedure in Section 4.2 with a grid search over group prior $\alpha$, choosing the head-to-tail ratio from the range {0, 0.004, 0.008, …, 0.996, 1.0}. We pick the parameter *combination* that yields the lowest balanced error on a held-out validation set
>
> We show below the **balanced error**  on the test and validation samples, with *lower* being *better* (the numbers are slightly different from the paper because they are evaluated on different validation-test splits):
> \begin{aligned}
> &~~~~\text{Test} &\text{Validation} \\\\
> \text{Chow's rule}~~~~ &~~~~ 0.512 \pm 0.004 &0.498 \pm 0.011 \\\\
> \text{Plug-in [Balanced, heuristic]}~~~~ &~~~~  0.291 \pm 0.008 &0.282 \pm 0.007\\\\
> \text{Plug-in [Balanced, grid search]}~~~~ &~~~~ 0.284 \pm 0.007 &0.264 \pm 0.008
> \end{aligned}
>
> Although the grid search provides gains on the validation set, on the test set, our heuristic approach to picking the multipliers and group priors yields metrics that are only slightly worse than grid-search. Moreover, the differences are within standard errors.
>
> > In the experimental section, only two compared methods are used. I am not sure if more compared methods are needed to validate the effectiveness of the proposal.
>
> To the best of our knowledge, there are **no prior L2R methods that can handle general (non-decomposable) evaluation metrics**. We therefore compare our approach against **representative** methods that minimize the standard classification error.
>
> The reason we do not pick an exhaustive list of methods from the L2R literature is because most of these methods **share the same optimal solution** (Equation 2), which as discussed in Section 3 (Remark 1), is notably different from the optimal solution for the balanced or worst-group errors.
>
> We believe the two baselines we compare against represent **two broad categories** of L2R approaches, namely post-hoc methods (represented by Chow’s rule), and loss-based methods (represented by by the CSS baseline of Mozannar & Sontag (2020)).

---

> > ### Comment · Reviewer_91tX · 2023-11-22
> >
> > Thanks for your reply! My concern is resolved and I am willing to increase my score.

---

### Official Review · Reviewer_RZqs · 2023-10-28

**Soundness:** 4 excellent
**Presentation:** 3 good
**Contribution:** 4 excellent
**Rating:** 8
**Confidence:** 3

**Summary:**

This paper considers the problem of classification with rejection when data distribution is long-tailed. In this case, metrics considering the data distribution such as balanced error is more appropriate than traditional misclassification error. For the case when learning a classifier and a separate rejector, authors derive corresponding Bayes optimal solutions for the problem setting, showing many existing methods are not optimal for metrics such as balanced error. Authors then propose a modification method based on the derived Bayes optimal solution, and empirically show its superiority.

**Strengths:**

- The paper considers an important problem setting combination, learning with rejection and class-imbalanced distribution.
- The analysis is rigorously conducted with high quality and significance.
- The paper is overall well structured and well written. It is easy and joyful to follow.
- The paper provide appropriate literature reivew for related work.

**Weaknesses:**

Limit of the proposed framwork is not properly addressed. For example, are there any popular metrics not covered?

**Questions:**

This paper considers the learning setting where a classifer and a separate rejector function exist. There is also another framework learning solely one classifier and later uses its output distribution over labels to conduct abstention. Is it impossible for this approach to derive Baye optimal solutions, or it is simply not the scope of the paper?

---

> ### Author Response · Authors · 2023-11-22
> **Response to Reviewer RZqs**
>
> We thank the reviewer for the encouraging comments.
>
> > are there any popular metrics not covered?
>
> We are currently able to minimize the **balanced error** and the **worst-group error**. Using techniques similar to Yang et al. (2020), our approach can be extended to handle metrics of the form  $\psi\left(e_1(h, r), \ldots, e_K(h, r)\right)$ in Secton 5, provided $\psi$ is either *convex* or *fractional-linear*. Metrics which do not fall under this family cannot be directly handle by our framework.
>
> > This paper considers the learning setting where a classifer and a separate rejector function exist. There is also another framework learning solely one classifier and later uses its output distribution over labels to conduct abstention. Is it impossible for this approach to derive Baye optimal solutions, or it is simply not the scope of the paper?
>
> Thanks for the question. If we understand, the reviewer is referring to settings where the rejector $r(x)$ is strongly coupled with the underlying classifier $h(x)$ or scorer $f(x)$, e.g., is given by the margin or softmax probability of the predicted label.
>
> One can in fact strengthen Equation 2 to derive the optimal choice of $r$ given a **fixed $h$**.  Our main results (e.g., Theorem 2) can similarly be strengthened, and cover the case where one does not jointly optimise over both $h$ and $r$. For a fixed classifier $h$, the optimal rejector in Theorem 2 will take the form:
>
> $$
> r\^\*(x) = 1 \~\~\\iff\~\~
> \frac{1}{\alpha^*_{[h(x)]}} \cdot \eta_{h(x)}(x)
> <
> \sum_{y' \in [L]}  \left( \frac{1}{\alpha^*_{ [y'] }}  - \mu^*_{ [y'] } \right) \cdot \eta_{y'}(x) - c
> $$
>
> Of course, if one does not choose $r$ in an optimal manner, this could lead to a globally sub-optimal solution. We can nonetheless add a discussion of this point.

---

> > ### Comment · Reviewer_RZqs · 2023-11-22
> >
> > Thank you for the clear response.
> >
> > My question was on the case of optimising strongly coupled $h(x)$ and $r(x)$ at the same time. I think this can be somehow covered by the main results of the manuscript. It is also nice to see the optimal rejector for a fixed $h(x)$, which may benefit some post hoc situations.

---

> ### Author Response · Authors · 2023-11-22
> **Response to follow-up question**
>
> This is indeed an interesting question! The reviewer is correct that our results can be extended to handle joint optimization of the classifier and rejector.
>
> One possible approach is to formulate the Lagrangian for the constrained optimization problem in Equation (9), and replace the indicators in the resulting Lagrangian objective with a surrogate loss function. We may consider the joint losses proposed recently by Mozannar & Sontag (2020) or Verma & Nalisnick (2022) as candidate surrogate losses. The resulting optimization would involve a maximization over the Lagrange multipliers and a minimization over the classifier-rejector parameters.
>
> We will be happy to include a discussion on this in the paper and mention it as a part of our future work.

---

> > ### Comment · Reviewer_RZqs · 2023-11-22
> >
> > Thank you for your prompt response. I am happy to be helpful on improving the manuscript.

---

> > > ### Author Response · Authors · 2023-11-22
> > > **Thank you!**
> > >
> > > Thank you, we appreciate the feedback and helpful comments!

---

### Official Review · Reviewer_iFdp · 2023-10-31

**Soundness:** 4 excellent
**Presentation:** 4 excellent
**Contribution:** 4 excellent
**Rating:** 8
**Confidence:** 4

**Summary:**

This paper derives and examines the Bayes optimal solution to minimizing the balanced error metric when a reject option is available (rejection is assumed to have a fixed cost). This problem is much more challenging than learning to reject for plain error because, as the authors show, the rejection thresholds for the various classes are coupled. Moreover, no reduction to cost sensitive learning with fixed per-class misclassification costs exist, since these costs change depending on the fraction of examples in each class not rejected.

Given the challenges, the authors present an approximate scheme for minimizing balanced-error with a reject option, and they compare its effectiveness to recent methods for learning to reject on 3 image classification datasets. The proposed method out-performs or matches the compared-to methods across all datasets.

Additionally, the authors formulate the Bayes optimal solution for minimum balanced error rate with rejection to one for any metric that is a function of the per-class error rates. This allows for minimizing the maximum per-class error rate, and the G-mean metric; as well as for balancing error rates across an arbitrary partition of classes -- e.g., balancing the error rate across the set of frequent classes and the tail classes.

**Strengths:**

This is the first paper to formulate the Bayes optimal solution for minimum balanced error classification with a rejection option, and to provide a means of optimizing it. The problem itself is rather significant since there are high stakes scenarios in which it's desirable for a classifier to perform equally well across groups, and for a rejection or abstention option to be available so that cases can be escalated to humans for more careful consideration.

The experiments reasonably demonstrate the method's effectiveness, and the authors generalize the form of the given solution to metrics other than balanced accuracy.

**Weaknesses:**

The major weakness of this paper is that the proposed optimization scheme is only approximate and an analysis of the gap is not given. Is it possible to say how much grid search over the space of lagrange multiplers is needed to find a good solution? Also, the proposed optimization scheme finds a fixed point of the alpha parameters for given values of lagrange multipliers. But is this fixed point unique? And if not, are they all equally good? These questions are not addressed in the paper.

**Questions:**

1) Is it possible to say how much grid search over the space of lagrange multiplers is needed to find a good solution?

2) the proposed optimization scheme finds a fixed point of the alpha parameters for given values of lagrange multipliers. But is this fixed point unique, and if not, are they all equally good?

---

> ### Author Response · Authors · 2023-11-22
> **Response to Reviewer iFdp**
>
> We thank the reviewer for the encouraging comments and questions.
>
> > Is it possible to say how much grid search over the space of lagrange multiplers is needed to find a good solution?
>
> Using the re-parametrization trick described in Appendix C.1, for $K$ groups, we only need to search over $K-1$ multiplier parameters. For our experiments, we consider settings with two groups (head and tail), requiring us to tune **only one** multiplier parameter. In our experiments, we find it sufficient to pick this parameter from a small set  {1, 6, 11}. We tried a grid search over a fine-grained range but found the gains to be small. Please feel free to also see response to Reviewer 91tX.
>
> > the proposed optimization scheme finds a fixed point of the alpha parameters for given values of lagrange multipliers. But is this fixed point unique, and if not, are they all equally good?
>
> The proposed optimization scheme was mainly intended as a heuristic to pick the group prior for the plug-in estimator. To confirm its efficacy, we repeated the CIFAR-100 experiments replacing our heuristic optimization scheme for picking $\alpha$ with a **fine-grained grid search**.  Specifically, we chose the head-to-tail ratio from the range  {0, 0.004, 0.008, …, 0.996, 1.0}
> and the multiplier from the wider range {0.25, 0.5, ..., 11} to yield the minimum balanced error on a held-out validation set.
>
> We show below the **balanced error**  on the test and validation samples, with *lower* being *better* (the numbers are slightly different from the paper because they are evaluated on different validation-test splits):
> \begin{aligned}
> &~~~~\text{Test} &\text{Validation} \\\\
> \text{Chow's rule}~~~~ &~~~~ 0.512 \pm 0.004 &0.498 \pm 0.011 \\\\
> \text{Plug-in [Balanced, heuristic]}~~~~ &~~~~  0.291 \pm 0.008 &0.282 \pm 0.007\\\\
> \text{Plug-in [Balanced, grid search]}~~~~ &~~~~ 0.284 \pm 0.007 &0.264 \pm 0.008
> \end{aligned}
>
> Although the grid search provides gains on the validation set, on the test set, our heuristic approach yields metrics that are only slightly worse than grid-search. Moreover, the differences are within standard errors.

---

> > ### Comment · Reviewer_iFdp · 2023-11-23
> > **Reply to authors response**
> >
> > I have read the authors' response to my review.
> >
> > Thank you for running the expanded grid search to further evaluate the power iteration-like heuristic. Adding these results to the appendix would be helpful.
> >
> > I will keep my review as is.

---

### Official Review · Reviewer_RJqn · 2023-11-02

**Soundness:** 3 good
**Presentation:** 3 good
**Contribution:** 3 good
**Rating:** 8
**Confidence:** 4

**Summary:**

The manuscript considers the problem of learning optimal classifiers with reject option, i.e., there's a certain budget, and a fixed cost, for refraining from making any predictions on test instances. The problem of 'learning to reject' has been studied before, but mostly in the setting when the performance metric of interest is classification accuracy (or 0-1 loss). In this work, the authors consider more general and practical performance measures, balanced error and worst-group error in particular. The authors derive Bayes optimal classifiers (and rejectors) for the general measures, and provide a practical algorithm for estimating them with samples.

Two key contributions: 1) Clear formulations of the learning to reject problem, various metrics of interest, and deriving Bayes optimal forms for the different metrics. While general performance measures have been considered before in different supervised learning settings, deriving general results for classifiers with reject option is novel as far as I can tell. 2) Algorithm for estimating the classifiers and rejectors in practice for general performance measures, and supportive empirical results on standard datasets.

Overall, I find the paper to be sufficiently interesting to any ML audience, with technical depth and contributions par for the venue. I've some questions for the authors listed below, but the paper is a clear accept for me.

**Strengths:**

As I mentioned in my summary the paper makes key contributions, and each of the contributions involves good technical rigor and presentation which I very much appreciate: 1) Clear formulations of the learning to reject problem, various metrics of interest, and deriving Bayes optimal forms for the different metrics. While general performance measures have been considered before in different supervised learning settings, deriving general results for classifiers with reject option is novel as far as I can tell. 2) Algorithm for estimating the classifiers and rejectors in practice for general performance measures, and supportive empirical results on standard datasets.

The paper is well-written, ideas flow coherently, related results are placed well in context, and it was a pleasure to read!

**Weaknesses:**

These are more nit picks than weaknesses.
1. Some intuitive justification for the Bayes optimal forms would have been useful. As it stands, the material is somewhat dense, and for a reader outside the area, some of these results might be confusing. For instance, I would love to know if there's an intuitive explanation for the 'discount' factor $-\mu^*_{[y']}$ in (2).
2. I found it a bit of a leap to go from deterministic classifiers in Theorem 2 to stochastic classifiers in Theorem 5. Balanced error does involve weights that are distribution dependent, while general $\psi$ metrics studied in Theorem 5 I believe are functions that do not depend on the distribution in any way other than via the arguments $e_j(h,r)$'s. So where is the stochasticity arising from? Also, it would be good to draw corollaries where for some simple forms of \psi functions, you indeed get deterministic classifiers and rejectors (when $h^{(1)} = h^{(2)}$ holds, etc.)

**Questions:**

I would like the authors to address the questions in the 'weaknessess' section in their rebuttal.

---

> ### Author Response · Authors · 2023-11-22
> **Response to Reviewer RJqn**
>
> We thank the reviewer for the encouraging and detailed comments.
>
> > Some intuitive justification for the Bayes optimal forms would have been useful. For instance, I would love to know if there's an intuitive explanation for the 'discount' factor
>
> Thanks for the suggestion. To build intuition for the Bayes optimal rejector, let’s consider a simpler risk with a fixed cost of $\beta_y$ for errors on class y:
>
> $\min_{h, r}~    \sum_i \beta_i \cdot \mathbb{P} \left( y = i, h(x) \ne y, r(x) = 0 \right)  + c \cdot \mathbb{P}\left( r( x ) = 1 \right)$
>
> The Bayes-optimal rejector for this risk is of the form:
>
> $$r\^\*(x) = 1 \~\~\\iff\~\~ \\max\_y \\beta\_y \\cdot \\eta_y(x) < \\sum\_i \\beta\_i \\cdot \\eta\_i(x) - c~~~~~~(i)$$
>
> One may consider the term $\max_y \beta_y  \cdot \eta_y(x)$ as a measure of classifier confidence, and the right-hand side as an instance dependent threshold. When $\beta_y = 1, \forall y$, Equation (i) is the same as Chow’s rule: $\~\~r\^\*(x) = 1 \~\~\\iff\~\~ \\max\_y \\eta_y(x) < 1 - c$.
>
> Note that the rejector $r^*$ is unconstrained in what rejection rate it has on individual classes. Suppose we now additionally constraint the rejector to satisfy a particular rejection rate for each class:
>
> $$\\min\_{h, \~r}\~    \sum_i \beta_i \cdot \mathbb{P} \left( y = i, h(x) \ne y, r(x) = 0 \right)  + c \cdot \mathbb{P}\left( r( x ) = 1 \right)$$
>
> $$\\text\{s.t.\}\~\~ \\mathbb\{P\}\\left( r( x ) = 1, y = i \\right) = B_i, \\forall i,$$
>
> for budgets $B_1, \ldots, B_L$.
>
> In this case, the optimal rejector (under additional distributional assumptions) takes the form:
> $$r\^\*(x) = 1 \~\~\\iff\~\~ \\max\_y \beta_y \cdot \eta_y(x) < \sum_i (\beta_i - \mu_i) \cdot \eta_i(x) - c~~~~~~(ii),$$
> where the discount factors $\mu_i$s ensure that the rejector satisfies the budget constraints. The optimal rejector again uses the term $\max_y \beta_y \cdot \eta_y(x)$ to measure confidence, but differs in how it thresholds the confidence measure. For example, in the case of binary labels, the rejector can be seen as applying a different constant threshold for each predicted class (see e.g. Corollary 3).
>
> We will be happy to add this discussion to the paper.
>
> > I found it a bit of a leap to go from deterministic classifiers in Theorem 2 to stochastic classifiers in Theorem 5.
>
> We would first like to clarify that the balanced error is a **special case** of the formulation in Equation 16 for $\psi(z_1, \ldots, z_K) = \frac{1}{K} \sum_k z_k$. In this case, $\psi$ happens to be a simple **linear** function, and the optimal solution is deterministic. When $\psi$ is a general **non-linear** function (e.g. the worst-group error), the optimal solution admits a stochastic form.
>
> The technical details of how we arrive at a stochastic solution are provided in Appendix B. The key point is that in order to solve Equation 16, we formulate an equivalent optimization problem over the **space of confusion matrices** $\mathcal{C}$ (see Appendix B.1 and Equation 28). In turn, attaining the optimal confusion matrix requires the use of randomization over $(h, r)$ pairs.
>
> When $\psi$ is linear, as is the case with the balanced error, the optimal solution is a **boundary point** in $\mathcal{C}$, and therefore a deterministic $(h, r)$ pair. When $\psi$ is non-linear, the optimal solution can be an **interior point**, and therefore require randomizing between two $(h, r)$ pairs.
>
> We will be happy to provide more details when transitioning from deterministic to stochastic rejectors.
>
> > Also, it would be good to draw corollaries where for some simple forms of \psi functions, you indeed get deterministic classifiers and rejectors
>
> Thanks for the suggestion. When $\psi(z_1, \ldots, z_K) = \sum_k \beta_k z_k$ is a linear function with coefficients $\beta_k \in \mathbb{R}$, the optimal solution is the deterministic pair $(h, r)$ given in Theorem 9 in the appendix. In this case, indeed $h^{(1)} = h^{(2)}$ and $r^{(1)} = r^{(2)}$. We will add a note on this in the paper.

---

> > ### Comment · Reviewer_RJqn · 2023-11-22
> > **Thanks!**
> >
> > Thank you for the clarifications. It would be helpful to include some of these in the main text.
> > I really like this work!

---

> > > ### Author Response · Authors · 2023-11-22
> > > **Thank you!**
> > >
> > > We will certainly include in the paper the additional clarifications we provide in the rebuttal. Thanks for the encouraging and helpful feedback.

---

### Comment · Area_Chair_7kSc · 2023-11-20
**Authors, please respond to the reviews**

Dear authors: Reviewers are generally positive about this submission, but they do have some concerns. Please submit your responses soon. Thank you!

---

> ### Author Response · Authors · 2023-11-20
> **Re: Authors, please respond to the reviews**
>
> Dear Area Chair,
>
> We appreciate the reminder. We are working on the rebuttal and will definitely submit it before the deadline.
>
> Thanks and regards,
>
> Authors

---

### Meta-Review · Area_Chair_7kSc · 2023-12-15

**Metareview:**

All reviewers are very positive about this submission. The contribution is a mathematically justified method to make classifiers with a reject option be approximately optimal for evaluation metrics that include balanced error and worst-group error.

One additional point that the authors could discuss is the relevance of the new method for fairness, such as when no subgroup of examples should have a higher error rate than others. The paper could also discuss previous non-obvious methods to optimize loss functions that involve interactions between classes, such as the approach in Optimal Thresholding of Classifiers to Maximize F1 Measure, ECML PKDD 2014.

**Justification For Why Not Higher Score:**

This paper could be called "traditional" in theoretical ML and will not be immediately impressive to a broad audience.

**Justification For Why Not Lower Score:**

The problem solved is important, and the solution is rigorous.

---

### Decision · Program_Chairs · 2024-01-16

Accept (spotlight)